# Sensory coding and the causal impact of mouse cortex in a visual decision

Peter Zatka-Haas[1,2†], Nicholas A Steinmetz[1†§*], Matteo Carandini[3‡], Kenneth D Harris[1‡*]

[1]UCL Queen Square Institute of Neurology, University College London, London, London, United Kingdom; [2]Department of Physiology, Anatomy & Genetics, University of Oxford, Oxford, United Kingdom; [3]UCL Institute of Ophthalmology, University College London, London, London, United Kingdom

**Abstract** Correlates of sensory stimuli and motor actions are found in multiple cortical areas, but such correlates do not indicate whether these areas are causally relevant to task performance. We trained mice to discriminate visual contrast and report their decision by steering a wheel. Widefield calcium imaging and Neuropixels recordings in cortex revealed stimulus-related activity in visual (VIS) and frontal (MOs) areas, and widespread movement-related activity across the whole dorsal cortex. Optogenetic inactivation biased choices only when targeted at VIS and MOs, proportionally to each site's encoding of the visual stimulus, and at times corresponding to peak stimulus decoding. A neurometric model based on summing and subtracting activity in VIS and MOs successfully described behavioral performance and predicted the effect of optogenetic inactivation. Thus, sensory signals localized in visual and frontal cortex play a causal role in task performance, while widespread dorsal cortical signals correlating with movement reflect processes that do not play a causal role.

**\*For correspondence:**
nick.steinmetz@gmail.com (NAS);
kenneth.harris@ucl.ac.uk (KDH)

[†]These authors contributed equally to this work
[‡]These authors also contributed equally to this work

**Present address:** [§]Department of Biological Structure, University of Washington, Seattle, WA, United States

## Introduction

When presented with multiple possible behavioral options, an animal must make use of sensory inputs to select one, if any, of the potential actions, and then execute this action. Neural correlates of sensation, action selection, and action execution have been found in multiple brain regions (*Freedman and Assad, 2006*; *Goard et al., 2016*; *Hernández et al., 2010*; *Hirokawa et al., 2019*; *Liu et al., 2013*; *Park et al., 2014*; *Pho et al., 2018*; *Platt and Glimcher, 1999*; *Raposo et al., 2014*; *Steinmetz et al., 2019*; *Wei et al., 2019*; *Yang et al., 2016*). Nevertheless, the presence of such correlates does not imply that these areas play a causal role in performing the task in question (*Hong et al., 2018*; *Katz et al., 2016*; *Kawai et al., 2015*).

A key confound when interpreting neural correlates of task variables is the widespread activity that pervades the brain before movements (*Ahrens et al., 2012*; *Allen et al., 2017*; *Musall et al., 2019*; *Salkoff et al., 2020*; *Steinmetz et al., 2019*; *Stringer et al., 2019*). The most common neural correlates of behavior appear to be with whether the animal will move at all, rather than which particular action will be chosen. For example, in a visual discrimination task, neurons in nearly all brain structures are modulated prior to upcoming actions, whereas neurons predicting which action will be chosen are rare and confined to specific regions such as basal ganglia, midbrain, and frontal cortex (*Steinmetz et al., 2019*).

It is unclear whether this widespread, non-specific pre-movement activity is necessary for performance of a task. Traditional theories would not predict that brain-wide activity is required for simple sensorimotor tasks. Indeed, cortical inactivation has little effect in some well-trained behaviors (*Kawai et al., 2015*; *Pinto et al., 2019*). However, in other simple tasks such as licking following a

whisker deflection, performance is impaired by inactivation of many structures, including medial prefrontal cortex and hippocampus (*Le Merre et al., 2018*).

Inactivating a brain region during a choice task could have three consequences. First, inactivation could bias which alternative is selected, implicating the inactivated neurons in sensory processing or action selection (*Erlich et al., 2015*; *Goard et al., 2016*; *Guo et al., 2014*; *Licata et al., 2017*; *Seidemann et al., 1998*; *Znamenskiy and Zador, 2013*). Second, inactivation might disrupt the execution of the chosen action, for example, delaying it or reducing its accuracy (*Guo et al., 2015*), but not affecting the probability of the two alternatives. This would implicate the inactivated neurons in action execution, but not action selection. Finally, inactivation might have no effects on the current trial, suggesting that the region's activity in the task reflects corollary discharge (feedback from other brain areas involved in action selection and execution; *Crapse and Sommer, 2008*), redundancy with other regions (*Li et al., 2016*), or cognitive processes that only causally affect subsequent trials (*Akrami et al., 2018*; *Lak et al., 2020*).

Here, we use widefield calcium imaging, electrophysiology, and optogenetic inactivation to reveal the roles of dorsal cortical regions in vision, action selection, and action execution. We trained mice in a two-alternative unforced choice discrimination task (*Burgess et al., 2017*) and found correlates of vision (regardless of behavioral output) in visual and frontal cortex, and correlates of action execution (initiation of any movement, regardless of stimulus and choice) everywhere, but strongest in primary motor and somatosensory cortex. Neural correlates of action selection – that is, neural activity predicting which choice would be made independent of the stimuli – were not found in widefield imaging and were revealed electrophysiologically only in rare neurons in frontal areas. The effect of optogenetically inactivating a region closely mirrored its encoding of sensory stimuli rather than its encoding of action execution or action selection. For instance, primary motor and somatosensory cortices had the strongest action execution correlates, but inactivating them had only minor effects on action execution. We constructed a model of choice based on cortical activity and found that it provided a parameter-free prediction of the effects of optogenetic inactivation. These results reveal that the local causal role of dorsal cortical regions in this task primarily reflects their representation of sensory stimuli, rather than action selection or execution.

## Results

### A visual discrimination task affording multiple types of error

We trained mice to perform a two-alternative unforced-choice visual discrimination task (*Figure 1*; *Burgess et al., 2017*). Mice were head-fixed with their forepaws controlling a steering wheel surrounded by three screens (*Figure 1a*). Each trial began after the wheel was held still for a minimum duration. Grating stimuli appeared in the left and right screens together with an auditory Go cue (*Figure 1b–d*). Mice were rewarded with water for rotating the wheel to bring the higher-contrast stimulus into the center (termed Left or Right choice trials, according to which side's stimulus is driven to the center), or for holding the wheel still for 1.5 s if no stimulus was present (termed NoGo trials). Unrewarded trials ended with a 1 s white noise burst. Mice became proficient in the task, achieving 86 ± 9% correct choices (mean ± sd) on trials with single high-contrast gratings, and the probability of their choices varied with the contrasts of the two gratings (*Figure 1e–h*). In the absence of visual stimuli, moreover, mice correctly held the wheel still (NoGo) in 54 ± 16% of the trials.

In this task design, mice could make multiple types of error (*Figure 1h*). When a stimulus was present, mice could err with a NoGo (a 'Miss' trial, 16 ± 13% of all trials where a stimulus was present on one side, mean ± sd across sessions) or with a wheel turn in the incorrect direction ('Incorrect choice,' 7 ± 4%). The probability of both types of error increased with decreasing stimulus contrast, reaching a maximum of 27 ± 21% Miss rate and 9 ± 8% incorrect rate for single low-contrast stimuli on one side (*Figure 1h*). These two types of errors permit us to separate the overall tendency to respond (Choice vs. Miss) from the accuracy of perceptual discrimination (Correct vs. Incorrect choices).

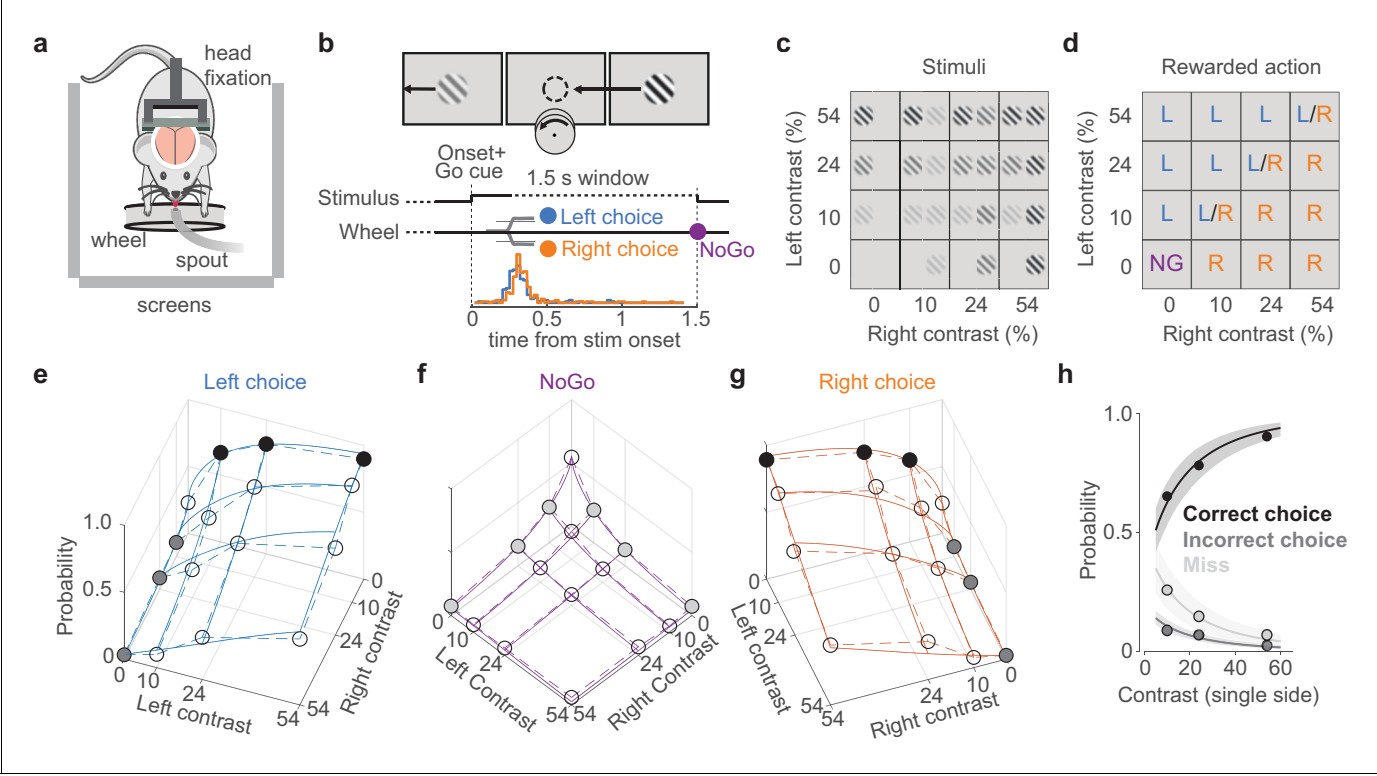

**Figure 1.** A visual discrimination task affording multiple types of error. (**a**) Behavioral setup, with the mouse surrounded by three screens. A spout delivers water rewards. (**b**) Example stimulus, with higher contrast on the right than left. The correct movement is to turn the wheel anticlockwise to bring the right stimulus to the middle (arrow and circle, not visible to the mouse). Middle: task timeline. Trials began after the mice held the wheel still for 0.2–0.6 s. Grating onset was accompanied by an auditory Go cue. Mice could then make a choice (Left/Right) or hold the wheel for 1.5 s (NoGo). Trials were separated by a 1 s interval. Bottom: distribution of movement onset times ('reaction time,' see Materials and methods) for correct Left (blue) and Right (orange) choices from one example mouse. (**c**) The 16 stimulus conditions. Gratings on each side could have one of four contrasts. (**d**) Rewarded actions depended on stimulus condition: Left (L), Right (R), NoGo (NG). When the contrasts were equal, Left and Right choices were rewarded randomly (L/R). (**e**) Probability of Left choices as a function of left and right contrast, averaged over 34 sessions in six mice (circles and dashed lines). Colored circles highlight unilateral stimulus conditions summarized in (**h**), where choices are Correct (black) or Incorrect (dark gray). Blue curves indicate the mean fit of a psychometric model fitted with hierarchical priors (see Materials and methods; *Burgess et al., 2017*). (**f, g**) Same format as in (**e**), showing NoGo trials and Right choice trials. Light gray dots indicate unilateral stimulus conditions where NoGo counts as a Miss. (**h**) Summary of these choices for unilateral stimulus conditions. Circles (with color fills as in **e–g**) show the probability of a Correct choice (black), Incorrect choice (dark gray), and incorrect NoGo (Miss; light gray). Curves and shaded regions are the mean and 95% credible intervals of the mean fit from the psychometric model.

## Activity related to stimuli and movement in the dorsal cortex

We measured cortical activity during task performance with widefield calcium imaging (*Allen et al., 2017*; *Berger et al., 2007*; *Chen et al., 2017*; *Makino et al., 2017*; *Wekselblatt et al., 2016*), which produces a signal tightly correlating with the summed spiking of neuronal populations at each cortical location (*Peters et al., 2021*). This revealed that the visual stimulus elicited a sequence of activations in the contralateral cortex, followed by widespread bilateral activity before a movement (*Figure 2a–e*). Activity began in the primary visual cortex (VISp) 47 ± 4 ms after stimulus onset (time to 30% of peak; median ± m.a.d. across six mice), spreading to secondary visual areas (e.g., VISal at 55 ± 8 ms), and then to frontal cortex (secondary motor area MOs at 98 ± 5 ms; *Figure 2—figure supplements 1* and *2*). The pattern of activity during these first 100 ms appeared to be independent of the trial's outcome of Go (Left or Right) or NoGo (*Figure 2b, c*). Later activity, however (150–200 ms following stimulus), strongly depended on whether the stimulus evoked a movement, with widespread bilateral activity seen on Go trials but absent on NoGo trials (*Figure 2b, c*, *Figure 2—figure supplement 1*, *Figure 2—videos 1 and 2*). This widespread activity preceded the mouse's movement, which did not occur until 229 ± 28 ms after stimulus onset (median ± m.a.d. across 39 sessions;

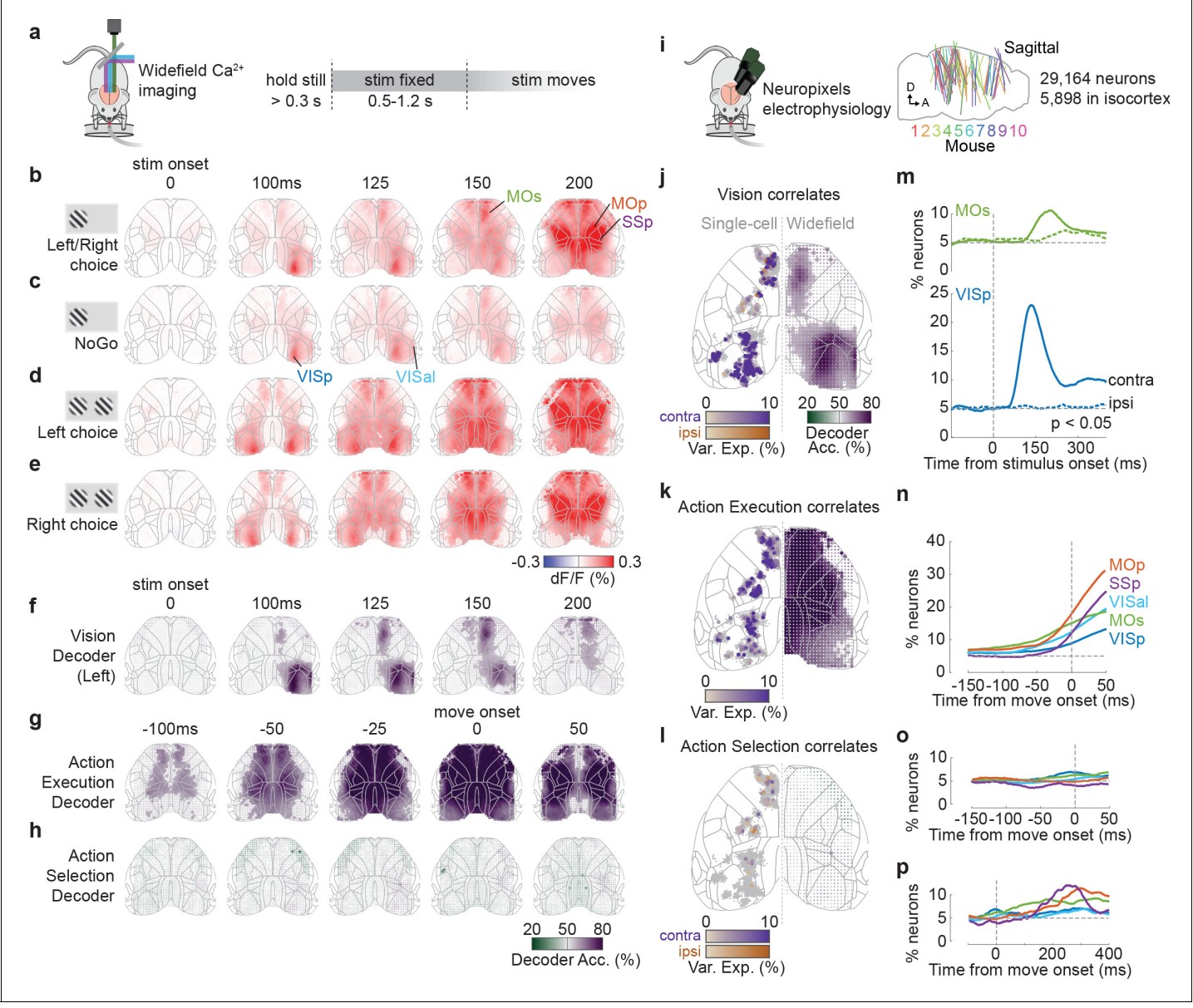

**Figure 2.** Activity related to stimuli and movement in the dorsal cortex. (a) Schematic of widefield calcium imaging. Mice expressed GCaMP6 in excitatory neurons (n = 7 mice) or all neurons (n = 2). Right: task timeline. Stimuli were fixed in place for 0.5–1.2 s to dissociate wheel turns from visual motion in recorded activity. (b) Cortical fluorescence (dF/F) for successive times relative to stimulus onset, averaged across 39 sessions in nine mice. Fluorescence is shown for trials with stimuli on the left (avg. 66% contrast) and a Left or Right choice. Gray lines: cortical areas defined from the Allen Common Coordinate Framework (CCF, image cropped to this region; *Wang et al., 2020*). The image is built of dots (one for each image pixel), whose shade reflects dF/F and whose size reflects statistical significance (larger dot: p<0.001, nested ANOVA). (c) Average dF/F for the same stimulus conditions as in (b) but for Miss (NoGo) trials. (d, e) Average dF/F for trials with equal non-zero contrast on both sides and a Left (d) or Right (e) choice. (f) Average Vision decoder accuracy, decoding whether a stimulus was present on the left screen from each pixel of the widefield image (see Materials and methods). Dot size reflects statistical significance as in (b). (g, h) Average Action Execution decoder accuracy (g; Go vs. NoGo) and Action selection decoder accuracy (h; Left vs. Right), aligned to movement onset. (i) Schematic of Neuropixels recordings. Right: sagittal map of probe insertion sites over 39 sessions in 10 mice (5898 cortical neurons). (j) Vision correlates. Comparison between contralateral Visual decoding maps obtained from widefield data (right hemisphere; 125 ms after stimulus onset) and a map of single neurons (left hemisphere jittered dots) that significantly encoded contralateral (purple) and ipsilateral (orange) stimuli based on kernel analysis. Dot brightness for neurons scales with the cross-validated variance explained of the stimulus kernel. Gray dots indicate neurons with no significant stimulus kernel. (k) Action Execution correlates. Same plotting scheme as in (j) but comparing widefield Action Execution decoding (Go vs. NoGo) at 25 ms before movement onset with the variance explained of a movement kernel. (l) Action Selection correlates. Same plotting scheme as in (j) but comparing widefield Action Selection decoding at 25 ms before movement onset with the variance explained of a choice kernel for contralateral (purple) and ipsilateral (orange) choices. (m) Proportion of VISp (blue)

*Figure 2 continued on next page*

*Figure 2 continued*

and MOs (green) neurons with significant contralateral (solid line) and ipsilateral (dashed colored line) stimulus decoding (p<0.05; binary decoder). Dashed gray lines: stimulus onset (vertical) and false-positive rate (horizontal). (**n**) Proportion of neurons that show significant Action Execution decoding (p<0.05; binary decoder) relative to movement onset. Colored lines indicate neurons grouped by CCF area. Vertical dashed line: movement onset. (**o**) Proportion of neurons with significant choice decoding (p<0.05; binary decoder). (**p**) Same as (**o**) but showing later times after movement onset.
© 2017, Steinmetz et al. https://creativecommons.org/licenses/bync/4.0/ Panel i adapted from *Steinmetz et al., 2017*, published under the Creative Commons Attribution - Non Commercial 4.0 International Public License (CC BY-NC 4.0).

The online version of this article includes the following video and figure supplement(s) for figure 2:

**Figure supplement 1.** Average cortical fluorescence (dF/F) for different task conditions.
**Figure supplement 2.** Widefield sequential activation.
**Figure supplement 3.** Neuropixels decoding analysis and encoding (kernel regression) analysis.
**Figure 2—video 1.** Widefield fluorescence in VISp, MOs, and MOp/SSp for different contralateral and ipsilateral contrast conditions.
https://elifesciences.org/articles/63163#fig2video1
**Figure 2—video 2.** Average widefield fluorescence in VISp, MOs, and MOp/SSp for different contralateral and ipsilateral contrast conditions, and choices.
https://elifesciences.org/articles/63163#fig2video2

reaction time defined as the earliest detected wheel movement after stimulus onset, see Materials and methods). However, while cortical activity was very different between Go and NoGo trials evoked by the same stimulus, cortical activity preceding Left vs. Right responses to the same stimulus appeared indistinguishable (*Figure 2d, e*).

The widefield signal was correlated with sensory and action execution variables, but not with action selection (*Figure 2f–h*). To show this, we used a binary decoder strategy that extends the choice probability method (*Steinmetz et al., 2019*; see Materials and methods). This method statistically tests whether one can decode a behavioral or stimulus variable from neural activity observed at a single place and time, while holding all other variables constant. We first asked which regions' activity could be used to decode the presence of a sensory stimulus on one screen, amongst trials with the same behavioral outcome and the same contrast on the other screen ('Vision' correlates). These Vision signals appeared in contralateral visual and frontal cortices (*Figure 2f*), starting in VISp at 50 ms and MOs at 100 ms (defined as the first time bin at which decoding performance significantly exceeded 50% across sessions; p<0.01, nested ANOVA). Further decoding performance was possible in ipsilateral MOs at 125 ms, but not elsewhere. We next asked which regions' activity could predict Go vs. NoGo choices, amongst trials with the same stimulus conditions ('Action Execution' correlates). This showed a different pattern: Go vs. NoGo could be decoded from cortical activity from 100 ms prior to movement onset, starting in primary motor and frontal cortex, and expanding to most imaged regions by 25 ms before movement (*Figure 2g*). Finally, we asked which regions could predict upcoming Left vs. Right choices amongst Go trials ('Action Selection' correlates) and found that none of the recorded regions supported this prediction (*Figure 2h*).

These imaging results were consistent with recordings from individual neurons (*Figure 2i–l*). We compared the widefield results to previous analyses of Neuropixels electrode recordings made in the same task (*Steinmetz et al., 2019*). To identify the correlates of individual neurons, we fitted single-neuron firing rates with an encoding model: a sum of kernel functions time-locked to stimulus presentation and to movement onset (*Figure 2i*; *Park et al., 2014*; *Steinmetz et al., 2019*). Consistent with the widefield imaging, neurons with significant stimulus encoding (see Materials and methods) were localized to contralateral VISp (25.2%, 255/1010 neurons; *Figure 2j*, *Figure 2—figure supplement 3*), contralateral MOs (5.6%, 59/1062), and rarely in ipsilateral MOs (1.4%, 15/1062). By contrast, neurons significantly predicting action execution were observed in broad swaths of cortex, including VISp (2.8%, 28/1010; *Figure 2k*), MOs (12.0%, 127/1062), primary motor cortex (MOp; 15.3%, 85/556), and primary somatosensory cortex (SSp; 20.1%, 62/308). Finally, neurons predicting which specific action would be selected were found above chance levels only rarely in MOs (2.3%, 24/1062), MOp (1.8%, 10/556), and SSp (3.0%, 9/308). Sensory signals were not only more common than action selection signals but also more lateralized. Indeed, the percentage of visually encoding neurons that encoded a contralateral stimulus was 100% in VISp and 80% in MOs. Conversely, neurons with preference for contraversive and ipsiversive actions were found approximately equally

(MOs: 1.3% [14/1062] contra, 0.9% [10/1062] ipsi; MOp: 0.9% [5/556] contra, 0.9% [5/556] ipsi; SSp: 2.6% [8/308] contra, 0.3% [1/308] ipsi; *Figure 2l*, *Figure 2—figure supplement 3*).

To determine the timing of vision, action execution, and action selection coding in the electrophysiologically recorded populations, we used the same binary decoding strategy that we had applied to widefield signals (*Figure 2m–p*, *Figure 2—figure supplement 3*; see Materials and methods). Stimulus decoding began at ~60 ms in contralateral VISp, ~110 ms in contralateral MOs, and ~180 ms in ipsilateral MOs (*Figure 2m*). Action execution decoding was first observed in each region 50–125 ms prior to movement onset (*Figure 2n*), while neurons predicting which action would be selected were rare prior to movement (*Figure 2o*) but increased substantially afterwards, reaching a peak 200–300 ms after movement onset (*Figure 2p*).

Data from widefield calcium imaging and Neuropixels recordings therefore consistently suggested that cortical correlates of vision and action execution were common, but correlates of action selection were rare. First, neurons in visual cortex VIS and frontal cortex MOs correlated with the sensory visual stimuli. Second, neurons in all recorded regions correlated with upcoming action, with primary motor area MOp showing the strongest action execution correlates. In contrast, and again consistent with the widefield data, neurons encoding the animals' selected action were rare.

## Optogenetic inactivation of sensory-coding regions biases choice

To assess whether these cortical correlates of stimulus and action execution are necessary for task performance, we used optogenetic inactivation. We inactivated each of the 52 sites across the dorsal cortex by shining a blue laser (1.5 mW) in transgenic mice expressing channelrhodopsin-2 (ChR2) in *Pvalb*-expressing inhibitory interneurons (*Olsen et al., 2012*), using transcranial laser scanning (*Guo et al., 2014*). Each randomly selected site was inactivated on ~1.4% of trials. Laser illumination started at the onset of visual stimuli and lasted until a choice (or NoGo) was registered. If the cortical correlates of sensory stimuli in VISp and MOs are necessary for performance, then inactivating these regions should bias the subject's choices away from contralateral stimuli. Likewise, if the non-specific bilateral correlates of action execution in other regions such as MOp are necessary for performance, then inactivating these regions should cause an increase in NoGo trials.

Inactivation of each cortical region affected Left/Right choices in direct proportion to the region's encoding of sensory stimuli (*Figure 3a–c*). We first focused on trials with equal contrast on both sides, to which animals responded with Left and Right choices (rewarded with 50% probability) and NoGo responses (Miss trials, unrewarded). Optogenetic inactivation affected the animals' choices when targeted at the visual cortex (VIS; light diffusion made it impossible to distinguish effects in individual visual areas *Guo et al., 2014*; *Li et al., 2019*) and at frontal cortex (MOs; *Figure 3a*), biasing choices away from stimuli contralateral to the inactivated hemisphere (−35%, p<0.0002 in VIS, −22%, p<0.0002 in MOs; permutation test; *Figure 3—figure supplement 1a–d*). The strength of the bias produced by inactivating a cortical location closely matched the accuracy with which that location's activity correlated with the presence of a contralateral stimulus (*Figure 3b, c*). The contralateral choices that were reduced by inactivation of VIS or MOs were replaced primarily by ipsilateral choices when bilateral visual stimuli were shown and by NoGos when only contralateral stimuli appeared (*Figure 3—figure supplement 1e, f*). Thus, when VIS or MOs were inactivated, the animal responded as if stimuli on the contralateral side was not there.

In contrast, inactivation caused no changes in the balance of Go/NoGo responses, no matter how strongly the inactivated region encoded upcoming action execution (*Figure 3d–f*). Inactivation of primary motor area MOp and primary somatosensory area SSp – whose activity most strongly differed between Go and NoGo trials – had no effect on the proportion of Go vs. NoGo choices (*Figure 3d*). We observed similar results across the cortex and there was no relationship between the strength with which a region's activity measured by widefield imaging predicted action execution (Go vs. NoGo choices) and the effect of inactivating that region on Go vs. NoGo behavior (*Figure 3e, f*). To verify that these differences did not reflect insufficient laser power, we performed further experiments at 1.5, 2.9, and 4.25 mW and found inactivation of MOp and SSp did not increase NoGo probability at any laser power (*Figure 3—figure supplement 1f*). Inactivation of MOp (but not SSp) did, however, reduce wheel peak velocity by a small amount for both Left and Right choices non-specifically (−21.2%, p<0.01, t-test; *Figure 3—figure supplement 2e*).

Consistent with a causal role in sensory encoding, visual and frontal cortex were most needed precisely at the time when they encoded sensory stimuli (*Figure 3g–i*). To determine the times at

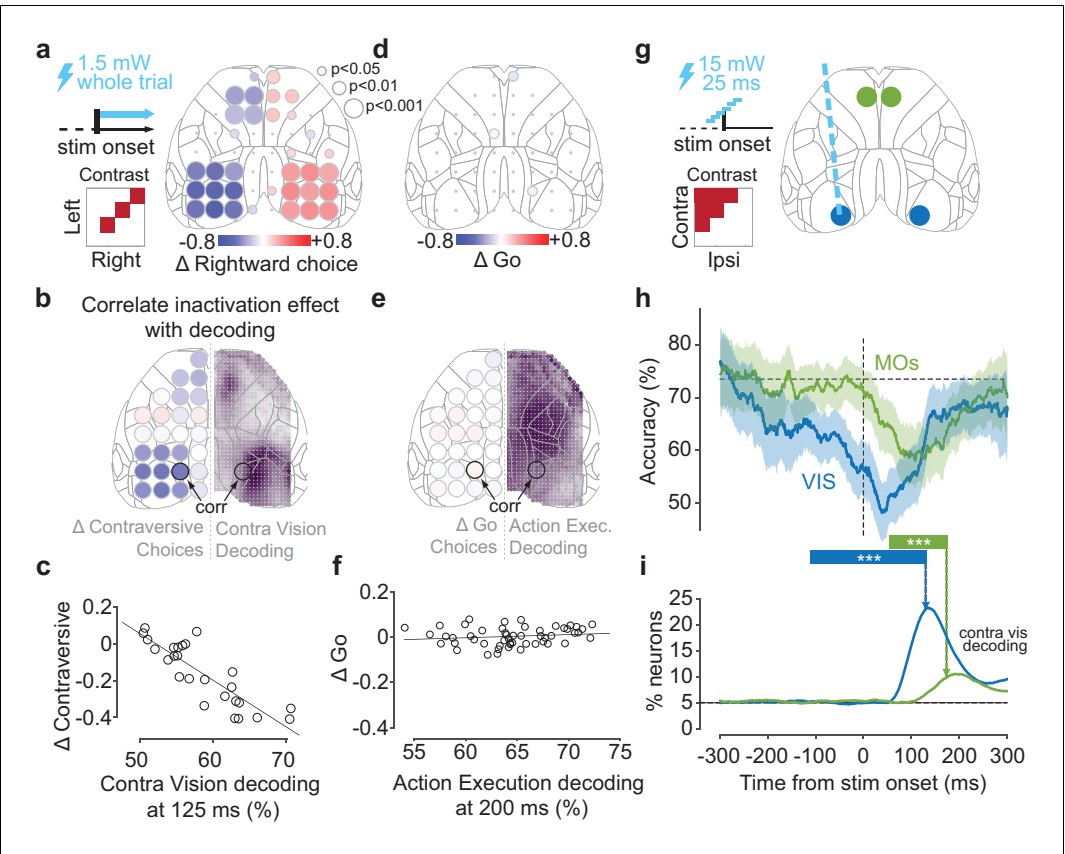

**Figure 3.** Optogenetic inactivation of sensory-coding regions biases choices. (a) Scanning inactivation in mice expressing channelrhodopsin-2 (ChR2) in parvalbumin-expressing inhibitory neurons (91 sessions in five mice; see Materials and methods). On ~75% of trials, a blue laser (1.5 mW, 40 Hz sine wave) illuminated one of 52 locations, from stimulus onset until a choice or NoGo was registered. Right: change in probability of rightward choices for 52 stimulation sites for trials with equal non-zero contrast on each side. Dot size indicates statistical significance (permutation test, see Materials and methods). (b) Comparison of the maps for contralateral Vision decoding (right hemisphere, from *Figure 2f* at 125 ms, dot size not scaled by significance) and for effect of inactivation (left hemisphere; from panel (a) combining across hemispheres). (c) Relationship between the choice bias caused by optogenetic inactivation across 26 sites, and the strength of contralateral stimulus decoding from widefield imaging at 125 ms post-stimulus, averaged over a region around the inactivated coordinate ('corr' in panel b). (d) Same as in (a) for the change in proportion of Go (Left or Right) choices for equal non-zero contrast trials. (e, f) Same as in (b, c), for the relationship between Action Execution (Go vs. NoGo) decoding at 200 ms after stimulus onset and the change in Go choices induced from inactivation. (g) Pulse inactivation experiments (65 sessions in six mice). On ~66% of trials, a 25 ms laser pulse was presented randomly between −300 and +300 ms relative to the stimulus onset (vertical dashed line), in one of four locations, chosen randomly (map on right). (h) Trial accuracy as a function of pulse time, averaged over trials where contralateral contrast > ipsilateral contrast. Blue curve: VIS inactivation; green curve: MOs' inactivation; black horizontal dashed line: mean accuracy in no-laser trials. Curves show the average pooled over all sessions and mice, smoothed with a 100 ms boxcar window. Shaded regions: 95% binomial confidence intervals. *** indicates the intervals in which accuracy differs significantly from control trials (p<0.0001; $\chi^2$ test). (i) Time course of contralateral Vision decoding (solid lines from *Figure 2m*). Vertical dashed arrows indicate the end of the critical intervals in (h). Black horizontal dashed line indicates chance level. The online version of this article includes the following figure supplement(s) for figure 3:

**Figure supplement 1.** 52-coordinate and mixed-power optogenetic inactivation.
**Figure supplement 2.** Effect of visual contrast and optogenetic inactivation on wheel movements.
**Figure supplement 3.** Pulsed inactivation and electrophysiological recording.

which activity in different regions played a causal role, we inactivated VIS and MOs with a brief laser pulse (25 ms, 15 mW) at different delays from stimulus onset (*Figure 3g*). Inactivation of VIS significantly impaired task performance around the time of stimulus onset (−110 to +130 ms; *Figure 3h*, *Figure 3—figure supplements 2g* and *3a, b*). Inactivation of MOs impaired performance at a later time window (+52 to +174 ms). These windows preceded the time of these mice's choices at 273 ± 21 ms (median ± m.a.d. across 65 sessions, on laser-off trials). Consistent with the scanning inactivation results, pulsed inactivation in MOp produced no significant performance impairment at any time (*Figure 3—figure supplements 2h* and *3a*). It might seem paradoxical that inactivation in VIS prior to stimulus onset could affect behavior; however, pulse activation of inhibitory cells suppresses cortical activity for over 100 ms (*Figure 3—figure supplement 3c, f*; *Olsen et al., 2012*). The critical time for inactivation is thus the last moment at which inactivation affects behavior, which corresponds to the time at which the suppressed region has completed its contribution to the task (this time will also be unaffected by the duration of the pulse-evoked suppression). Notably, this critical time for VIS and MOs mirrored the time of peak contralateral stimulus decoding in these regions as previously identified in electrophysiological recordings (*Figure 3i*).

## A lawful relationship between cortical activity and decisions predicts the effects of inactivation

To formally relate neural activity to the mouse's behavior, we constructed a neurometric model, which predicts actions and choices based on neural activity. In contrast to the analyses above (*Figure 3*), which focused on specific stimulus conditions, this model summarizes how cortical activity shapes the animals' decisions for all contrast combinations.

In the model, a hypothesized subcortical circuit computes a weighted sum of population activity of visual and frontal cortex to compute the evidence for a choice on each side (*Figure 4a*). Because only the visual and frontal cortex had a locally causal role in choice (*Figure 3*), we defined the decision variables as a weighted sum of activity in VISp and MOs, obtained from widefield imaging at the time this activity is causally relevant. The inputs to the model are thus the activities $V_L$ and $V_R$ of the left and right visual cortex (VISp) at time 75–125 ms after stimulus onset, and the activities $M_L$ and $M_R$ of the left and right frontal cortex (MOs) at time 125–175 ms. A weighted sum of these four variables yields the 'decision variables' representing the evidence for choices on the left and on the right:

$$Z_L = \alpha_L + v_c V_R + v_i V_L + m_c M_R + m_i M_L$$

$$Z_R = \alpha_R + v_i V_R + v_c V_L + m_i M_R + m_c M_L$$

where $v_c$ and $m_c$ represent weights from contralateral visual and frontal cortex, $v_i$ and $m_i$ represent weights from ipsilateral visual and frontal cortex, and $\alpha_L$ and $\alpha_R$ are constant intercept terms. The two decision variables determine the animal's behavior probabilistically, using a three-way logistic ('softmax') function, where the probabilities of Left, Right, and NoGo responses are

$$P(Left)/P(NoGo) = exp(Z_L)$$

$$P(Right)/P(NoGo) = exp(Z_R)$$

$$P(NoGo) = 1 - P(Left) - P(Right)$$

We fitted the parameters of this model using the data from widefield imaging, keeping the data from inactivations aside. To determine the activity variables, we calibrated widefield recordings against electrophysiologically measured firing rates to provide an estimate of firing rate from the widefield fluorescence signal (see Materials and methods; *Figure 4—figure supplement 1*). To fit the weights, we used a hierarchical Bayesian approach that allowed for variability in weights between subjects and recording sessions (Materials and methods). The inactivation data was not used to constrain these weights, allowing us to use these data to later validate the model.

The estimated weights showed a fundamentally different structure for visual and frontal cortex (*Figure 4b*). The weights on VISp activity were subtractive ($v_c$ positive and $v_i$ negative): activity on

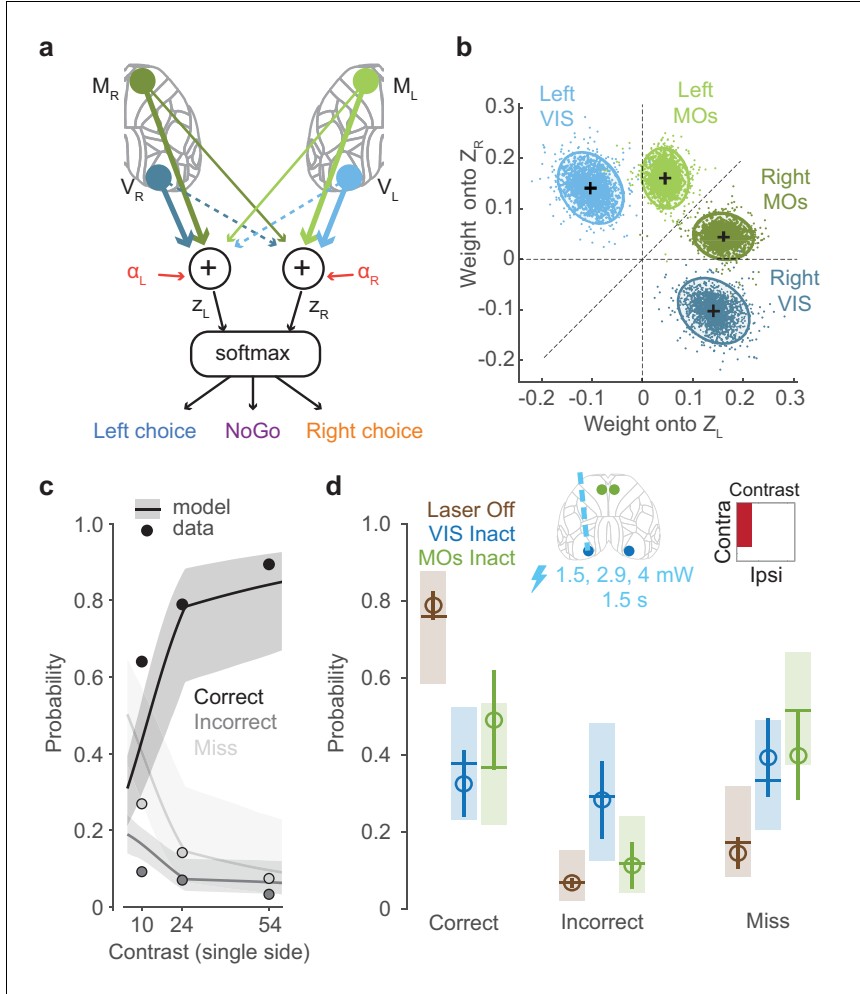

**Figure 4.** A lawful relationship between cortical activity and decisions predicts the effects of inactivation. (a) Schematic of the neurometric model. Activity in VISp and MOs is measured by widefield calcium imaging and used to estimate population firing rates on each trial (*Figure 4—figure supplement 1*). A weighted sum of activity in both hemispheres determines decision variables $Z_L$ and $Z_R$, and a softmax function generates the probability of each behavioral choice. (b) Posterior distributions of mouse-averaged model weights from the four regions to the two decision variables. Dots are draws from the posterior distribution of weights (see Materials and methods). Ellipses show 95th percentile of the posterior draws. Diagonal line illustrates the model weight symmetry. (c) Fit of the neurometric model to mouse choices (34 sessions in six mice, non-laser trials), using the same plotting scheme as in *Figure 1h*. (d) Model predicts effects of inactivation on Correct, Incorrect, and Miss trials, averaging over unilateral stimulus conditions. Open circles show the empirical probability for trials with laser off (brown), and for trials with optogenetic inactivation (1.5 s duration, multiple laser powers; see Materials and methods) of VIS (blue) and MOs (green), averaged across sessions (34 sessions in six mice). Error bars indicate the 95% confidence interval of the average estimate. Brown horizontal lines indicate the neurometric model fit to the average contrast value (~30%). Blue and green horizontal lines indicate mean predictions from the neurometric model, obtained by setting the activity of VISp (blue) and MOs (green) in the model to zero. Shaded regions indicate the 95% credible intervals of the model predictions.

The online version of this article includes the following figure supplement(s) for figure 4:

**Figure supplement 1.** Calibrating widefield activity to spike rate.

**Figure supplement 2.** Neurometric model fit and prediction.

one side of the visual cortex promoted contraversive choices and suppressed ipsiversive choices. By contrast, the weights on MOs activity were additive ($m_c > m_i$, but both positive): activity on one side of the frontal cortex promoted both contraversive and ipsiversive choices, but favoring the contraversive side. Thus, the decision variables represented a subtraction of visual cortical activity between

the two hemispheres, and an addition of frontal activity from both hemispheres. This difference may reflect the fact that 100% of stimulus-encoding neurons in VISp respond to contralateral stimuli, while 80% do so in MOs.

The model captured the behavior of the mice not only during imaging sessions (which were used to fit the model) but also during inactivation sessions, predicting the effects of inactivation even though inactivation data were not used in fitting model parameters (*Figure 4c, d*). During inactivation sessions, the model was able to capture the overall dependence on stimulus contrast of each type of choice: Correct choices, Incorrect choices, and Misses (non-laser trials; *Figure 4c*, *Figure 4—figure supplement 2*). To predict the results of optogenetic inactivation, we computed the model's prediction after setting the activity in a selected cortical region to zero. The model predicted a reduction in Correct choices and increase in Miss errors on inactivation of either VISp or MOs contralateral to a stimulus (*Figure 4d*).

Intriguingly, however, the model's predictions for VISp and MOs inactivation were not identical: due to the subtractive nature of the VISp weights, the model predicted that VISp inactivation should increase Incorrect choices, but that MOs inactivation should not. To test these predictions, we performed an additional optogenetic inactivation experiment in VIS and MOs (1.5 s duration, multiple laser powers; see Materials and methods). The empirical observations in the proportion of Correct, Incorrect, and Miss rates resembled the predictions from the neurometric model for VIS and MOs inactivation (*Figure 4d*, *Figure 4—figure supplement 2*). The neurometric model therefore provides a quantitative link between the activity observed in visual and frontal regions, the choices of the animal, and the effects of inactivation of those regions.

## Discussion

Our results reveal a tight link between the behavior of a mouse in a visual discrimination task and the sensory, but not the motor, correlates of the dorsal cortex. Visual stimuli elicited localized activity in the visual cortex (VIS) followed by frontal cortex (MOs). Activity all over the dorsal cortex – particularly in primary motor cortex (MOp) – predicted action execution, rising before movements in either direction. However, only a few neurons in frontal cortex showed correlates of action selection, predicting which direction the wheel would turn. Inactivation of VIS and MOs biased choices away from the contralateral stimulus, but inactivating MOp barely had any effect; across the cortex, the behavioral effects of local inactivation scaled linearly with the degree to which the inactivated region's activity correlated with sensory information. Moreover, the behavioral effects of inactivation matched the predictions of a neurometric model trained to compute choice based on cortical activity. These results suggest that the main contribution of dorsal cortex in this task is that VIS and MOs transfer sensory information to downstream circuits that select the animal's choice. Other areas such as MOp make a minor contribution to movement execution in this task.

Our data confirm the critical role of rodent frontal cortex (MOs) in sensory decisions but indicate that at least in this task MOs' role is primarily to route sensory information to other circuits, which feed it back a decision only once the movement is underway. MOs is thought to be critical for decisions based on sensory input (*Barthas and Kwan, 2017*), from diverse modalities, such as somatosensation (*Guo et al., 2014*; *Li et al., 2015*), audition (*Erlich et al., 2015*; *Siniscalchi et al., 2016*), olfaction (*Allen et al., 2017*; *Wu et al., 2020*), and vision (*Orsolic et al., 2021*). Our data are consistent with this hypothesis and suggest a primarily sensory role in the present task. Widefield imaging revealed no difference in MOs activity between trials of identical stimulus conditions but different choices (Left or Right). Electrophysiology indicated that single neurons that could predict Left vs. Right choices prior to movement onset were much rarer than neurons carrying a sensory signal, although the selected action could be decoded from more MOs neurons after the action had started. While inactivation of MOs could in principle affect behavior through off-target effects in VISp (*Otchy et al., 2015*), this is unlikely because MOs' inactivation maximally impairs behavior ~50 ms after the time when inactivation of VISp ceases to have any effect.

The rarity of pre-movement coding for the direction of the upcoming wheel turn in dorsal cortex suggests that in this task, actions are selected by subcortical circuits. Our neurometric model accurately predicted the subject's average choices based on cortical activity, but it assumed that the choice itself was determined stochastically by a circuit downstream of dorsal cortex. Indeed, previous electrophysiological recordings in the same task identified neurons with strong action selection

correlates in a variety of subcortical structures, including basal ganglia, superior colliculus, and zona incerta (*Steinmetz et al., 2019*). (Although cortical areas not accessible to widefield imaging could also in principle carry strong choice signals, previous electrophysiology did not find choice signals in any recorded non-dorsal cortical areas; *Steinmetz et al., 2019*.) The subcortical circuits carrying choice signals feed back to cortex, so it is certainly possible that a small number of cortical neurons participate in a distributed recurrent circuit that selects the animal's action. However, even in MOs, neurons predicting the selected action were sparse prior to movement onset and only became widespread after the action had already started. Thus, if MOs does play a role in selecting the subject's action, the neurons participating in this function would be substantially fewer in number than the neurons encoding the sensory stimulus. Moreover, while our data show that VIS plays an important role in routing sensory information to decision structures, they do not constrain whether this information flows directly from VIS to subcortical targets or indirectly via MOs.

The strongest and most widespread cortical correlate of behavior was with action execution (Go vs. NoGo), a correlate that may reflect a corollary discharge or the generation of movements irrelevant to task performance. For instance, activity in primary motor cortex MOp strongly differentiated between Go and NoGo trials, but inactivating MOp had no effect on the fraction of Go choices, causing only a slight decrease in wheel velocity. Therefore, these strong local action correlates are dispensable for the decision to choose Go over NoGo. Our data are thus consistent with previous studies showing that neural circuits exhibiting correlates of choice or action are not always necessary for it (*Erlich et al., 2015*; *Katz et al., 2016*; *Kawai et al., 2015*). Nevertheless, our results cannot rule out that unilateral MOp/SSp inactivation engages mechanisms that compensate for its effect on choices (*Li et al., 2016*), or that a distributed cortical representation of action is necessary for movement in this task, and our localized inactivations did not sufficiently disrupt this distributed representation. Further studies incorporating simultaneous multi-region inactivation would be required to test this.

What then might be the function of the strong MOp/SSp activity observed prior to action execution? Several possibilities remain. First, this activity might contribute to finely detailed execution of the ongoing movement. Indeed, some studies have demonstrated a causal role for MOp areas in dexterous fine forepaw movements (*Guo et al., 2015*; *Harrison et al., 2012*; *Sauerbrei et al., 2020*). It is possible that inactivation of MOp/SSp would perturb these fine details of paw configuration in a manner that our current videographic methods cannot detect, while sparing the control of more proximal muscles that suffice to turn the wheel. This possibility is supported by the effects of MOp inactivation on peak wheel velocity. Second, these regions might contribute to movements irrelevant to task performance, such as postural adjustments, fidgeting, or whisking. Third, they may reflect efference copy or corollary discharge (*Crapse and Sommer, 2008*; *Kaplan and Zimmer, 2020*) from other circuits involved in producing the choice. The function of such corollary discharge is unclear; however, it is not restricted to choice tasks as even spontaneous movements increase neuronal activity across the brain (*Stringer et al., 2019*). We speculate that this increased activity might serve to improve neural coding capacity throughout the brain at times of ongoing actions by boosting spike counts and lowering time constants (*Destexhe et al., 2003*).

The limited correlates of action selection we observed in this task stand in contrast to several previous studies, which showed choice coding in neurons of multiple cortical and subcortical regions (*Britten et al., 1996*; *de Lafuente and Romo, 2006*; *Goard et al., 2016*; *Guo et al., 2014*; *Li et al., 2015*; *Liu et al., 2013*; *Nienborg and Cumming, 2009*; *Wu et al., 2020*). There are two main hypotheses for how neural circuits develop such signals (*Nienborg and Cumming, 2010*). According to the 'bottom-up' hypothesis (*Britten et al., 1996*), neural noise leads to trial-to-trail fluctuations in sensory coding that impacts downstream decisions. According to the 'top-down' hypothesis (*Nienborg and Cumming, 2009*), choice probabilities reflect feedback of decisions made elsewhere in the brain. Evidence for the top-down hypothesis comes from the fact that choice probabilities grow stronger for several hundred ms after stimulus onset, and the fact that they are often stronger in higher-order cortex (*de Lafuente and Romo, 2006*; *Nienborg and Cumming, 2009*). Our data are consistent with studies that found essentially no choice probability in primary sensory cortex at least during the initial response phase (*Romo and de Lafuente, 2013*; *Sachidhanandam et al., 2013*). However, we also saw substantially less choice probability in frontal cortex than several previous studies (*Goard et al., 2016*; *Guo et al., 2014*; *Li et al., 2015*). We suggest two reasons for this apparent discrepancy.

First, many of the previous tasks that found widespread choice correlates were Go/NoGo tasks (*Allen et al., 2019*; *Allen et al., 2017*; *Lee et al., 2020*; *Poort et al., 2015*; *Puścian et al., 2020*; *Tang and Higley, 2020*; *Yang et al., 2016*). Our data are in fact entirely consistent with these results. We observed neural activity predicting whether a trial would be Go or NoGo in all regions examined, consistent with reports of Go/NoGo correlates in sensory cortex, frontal cortex, or anywhere else. Indeed, even outside of a task context neural correlates of motion are found all over the brain (*Stringer et al., 2019*). However, we found that blocking this activity in any region that did not also encode sensory information had had at most a minor role in action execution, suggesting that it might reflect processes such as corollary discharge.

Second, other studies using two-alternative choice tasks have shown larger numbers of neurons with action selection correlates than we observed in frontal, parietal, or sensory association cortex (*Duan et al., 2021*; *Erlich et al., 2011*; *Funahashi et al., 1991*; *Gu et al., 2007*; *Guo et al., 2015*; *Harvey et al., 2012*; *Li et al., 2015*; *Romo and de Lafuente, 2013*). To the best of our knowledge, all such tasks either used stimuli requiring integrating evidence over time or enforced a delay between stimulus onset and motor action, during which the selectivity for different actions grows steadily over several hundred ms. In our results, left/right action selectivity also grows for a few hundred ms after stimulus onset, but because our task does not require an evidence integration or delay period, the mice are by this time already executing their chosen action. Therefore, the encoding of ongoing actions we observe in MOs may be homologous to the encoding of upcoming actions at a similar time after stimulus onset in integration or delay tasks. In delay tasks, choice-related frontal activity is causally required to execute the actions (*Li et al., 2016*). In our task, however, the primary contribution of MOs seems to be to encode sensory activity: the only MOs neurons that could causally contribute to action selection are the few that encode the selected action before it is executed. We speculate that MOs may have different functions in different tasks, including relaying of sensory information to subcortical decision structures in our task and maintenance of short-term memories in delay tasks. This possibility is supported by the fact that MOs' representations of task-relevant stimuli develop over training (*Orsolic et al., 2021*; *Peters et al., 2021*).

It is conceivable that the small number of MOs neurons carrying decision signals in our task belong to a specific cellular subclass. For example, layer 5 pyramidal neurons (which dominate both extracellular and widefield calcium signals; *Peters et al., 2021*; *Sakata and Harris, 2009*) can be divided into two primary subclasses. The great majority are intratelencephalic (IT) neurons, which project bilaterally to cortex and striatum; a less numerous subclass of pyramidal tract (PT) neurons project primarily unilaterally to midbrain, pons, and medulla. PT neurons have distinct behavioral correlates in multiple tasks. In visual eyeblink conditioning, behavioral correlates emerge after training in V1 PT but not IT cells, and inhibiting these cells reduces behavioral responses (*Tang and Higley, 2020*). In a somatosensory two-alternative choice task, around two-thirds of PT neurons preferentially fired before contralateral licking responses, while IT neurons showed no lateral bias (*Li et al., 2015*). It is conceivable that the small number of MOs neurons with action selection correlates in our task might be PT cells; alternatively, action selection correlates might be found in rare IT and PT cells, but with lateral bias in only the PT cells, accounting for the partially lateralized effect of MOs' inactivation as has also been seen in licking behaviors (*Guo et al., 2014*).

In conclusion, the dorsal cortex's role in this task appears to be in relaying sensory evidence to a downstream circuit, which then determines behavior based on this evidence. While activity all over cortex reflects the occurrence of an impending action, this activity is not locally causal. Instead, the causal necessity of a cortical region depends on whether that region's activity encodes sensory stimuli. Modeling suggests that a stochastic circuit downstream of cortex could linearly weight activity in VIS and MOs to produce the subject's choice. The location of this downstream circuit is currently unconstrained, but may include subcortical structures with choice correlates such as basal ganglia and midbrain, as well as the rare neurons in frontal cortex that encode choice. Testing the causal relevance of these circuits will require further targeted inactivation experiments.

## Materials and methods

All experimental procedures were conducted at UCL according to the UK Animals Scientific Procedures Act (1986) and under personal and project licenses granted by the Home Office following appropriate ethics review.

## Mouse transgenic lines

For the widefield calcium imaging, we used transgenic mice expressing GCaMP6s in excitatory neurons (tetO-GCaMP6s [Jax #024742, RRID:IMSR_JAX: 024742] × Camk2a-tTA [Jax #007004, RRID: IMSR_JAX: 007004]), GCaMP6f in excitatory neurons (Ai95 [Jax #024105, RRID:IMSR_JAX: 024105] ×x Vglut1-IRES2-Cre [Jax#023527, RRID:IMSR_JAX: 023527]), or GCaMP6s in all neurons (Snap25-GCaMP6s [Jax #025111, RRID:IMSR_JAX: 025111]). For the optogenetic inactivation experiments, we used transgenic mice expressing ChR2 in parvalbumin-positive inhibitory interneurons (Ai32 [Jax #012569, RRID:IMSR_JAX: 012569] × Pvalb-Cre [Jax #008069, RRID:IMSR_JAX: 008069]). For the Neuropixels electrophysiology experiments, mice of multiple genotypes were used (*Steinmetz et al., 2019*). All mice were 10–73 weeks of age at the time of data collection. Mouse genotype and session counts for each figure are detailed in *Supplementary file 1*. Sample sizes were chosen based on previous studies using mice in visual decision tasks (*Burgess et al., 2017*; *Steinmetz et al., 2019*).

## Surgery

For widefield imaging, optogenetic inactivation, and electrophysiological recording experiments, mice were prepared with a clear skull cap similar to that of *Guo et al., 2014* and described previously (*Burgess et al., 2017*; *Steinmetz et al., 2019*). The implantation surgery proceeded as follows. The dorsal surface of the skull was cleared of skin and periosteum, and the junction between cut skin and skull was sealed with cyanoacrylate. The exposed skull was prepared with a brief application of green activator to ensure strong connection between cement and bone (Super-Bond C and B, Sun Medical Co, Ltd, Japan). The junction between skin and skull was again covered using dental cement (Super-Bond C and B). In most cases, a 3D printed 'cone' was attached to the head with cyanoacrylate and dental cement at this stage, surrounding the exposed skull and providing light isolation. A thin layer of cyanoacrylate was applied to the skull and allowed to dry. Two to four thin layers of UV-curing optical glue (Norland Optical Adhesives #81, Norland Products Inc, Cranbury, NJ; from Thor-Labs) were applied to the skull and cured (~10 s per layer) until the exposed skull was covered (thin layers were used to prevent excessive heat production). A head-plate was attached to the skull over the interparietal bone with SuperBond polymer.

## Behavioral task

### Apparatus

The two-alternative unforced choice task design was described previously (*Burgess et al., 2017*). In this task, mice were seated on a plastic apparatus with forepaws on a rotating wheel and were surrounded by three computer screens (Adafruit, LP097QX1) at right angles covering 270 × 70 degrees of visual angle (d.v.a.). Each screen was ~11 cm from the mouse's eyes at its nearest point and refreshed at 60 Hz. The screens were fitted with Fresnel lenses (Wuxi Bohai Optics, BHPA220-2-5) to ameliorate reductions in luminance and contrast at larger viewing angles near their edges, and these lenses were coated with scattering window film ('frostbite,' The Window Film Company) to reduce reflections. The wheel was a ridged rubber Lego wheel affixed to a rotary encoder (Kubler 05.2400.1122.0360). A plastic tube for delivery of water rewards was placed near the subject's mouth. For full details of the experimental apparatus including detailed parts list, see http://www.ucl.ac.uk/cortexlab/tools/wheel. All behavioral experiments were run using MATLAB Rigbox (*Bhagat et al., 2020*).

### Pre-stimulus quiescence

For three experiments, trials began after a period of no wheel movement (widefield imaging: 0.3–0.7 s, 52-coordinate inactivation experiment: 0.2–0.6 s, Neuropixels electrophysiology 0.2–0.5 s). For all other behavioral sessions, there was no constraint; however, trials were excluded post-hoc if wheel movement was detected −0.15 to +0.05 s from stimulus onset.

### Stimulus onset

At trial initiation, a visual stimulus was presented on the left, right, both, or neither screen. The stimulus was a Gabor patch with orientation 45°, sigma 9 d.v.a., and spatial frequency 0.1 cycles/degree. The grating stimuli on the left and right screens displayed at all combinations of 4 contrast levels,

totaling 16 contrast conditions. The proportion of trials of each stimulus type were weighted towards easy trials (high contrast vs. zero, high vs. low, medium vs. zero, and no-stimulus trials) to encourage high overall reward rates and sustained motivation. Zero contrast trials made up ~31% of trials; high contrast and medium contrast single-side contrast trials made up ~24%; equal contrast trials made up ~12% of trials; all other comparison contrast trials uniformly comprised the remaining trial types. For all experiments except for widefield imaging and Neuropixels electrophysiology (see 'Open-loop period' below), the onset of the visual stimulus also coincides with the onset of an auditory 'go cue' (12 kHz tone, 100 ms duration), marking the time at which the mouse can respond.

## Wheel movements

Wheel turns in which the top surface of the wheel was moved to the subject's right led to rightward movements of stimuli on the screen, that is, a stimulus on the subject's left moved towards the central screen. Put another way, clockwise turns of the wheel, from the perspective of the mouse, led to clockwise movement of the stimuli around the subject. A left or right choice was registered when the wheel was turned by an amount sufficient to move the visual stimuli by 90 d.v.a. in either direction. Movement onset time ('reaction time') is defined as time of the earliest detected wheel movement using the findWheelMoves3 algorithm (*Steinmetz et al., 2019*; https://github.com/cortex-lab/wheelAnalysis/blob/master/+wheel/findWheelMoves3.m; copy archived at swh:1:rev:8e1bc41791e3ea2debafadb0949be0c4e8ada3a8).

When at least one stimulus was presented, the subject was rewarded for driving the higher contrast visual stimulus to the central screen (if both stimuli had equal contrast, Left/Right choices were rewarded with 50% probability). When no stimuli were presented, the subject was rewarded if no turn (NoGo) was registered during the 1.5 s following the go cue.

## Open-loop period

For widefield calcium imaging and Neuropixels electrophysiology sessions, after stimulus onset there was a random delay of 0.5–1.2 s, during which time the subject could turn the wheel without penalty, but visual stimuli were locked in place and rewards could not be earned. The subjects nevertheless typically responded immediately to the stimulus onset, and trials were excluded if the initial wheel movement onset time was greater than 0.5 s. At the end of the delay interval, an auditory go cue was delivered (8 kHz pure tone for 0.2 s) after which the visual stimulus position became coupled to movements of the wheel and a choice could be made. Initial wheel turns were nearly always in the same direction as the final choice (96.6 ± 3.4%, mean ± sd across 39 sessions) indicating rare changes of mind (*Resulaj et al., 2009*) during the open-loop period. This small task modification was important to ensure that visual stimulus-related cortical activity was not inter-mixed with activity related to the auditory go cue and that movement-related activity was not inter-mixed with signals related to visual motion of the stimulus on the screen.

## Feedback

Immediately following registration of a choice or expiry of the 1.5 s window, feedback was delivered. If correct, feedback was a water reward (0.7–2.5 µL) delivered by the opening of a valve on the water tube for a calibrated duration. If incorrect, feedback was a white noise sound played for 1 s. During the 1 s feedback period, the visual stimulus remained on the screen. After a subsequent inter-trial interval of 1 s (or 2 s for the 52-coordinate inactivation experiment), the mouse could initiate another trial by again holding the wheel still for the prescribed duration.

## Training

Mice were trained on this task with the following shaping protocol. First, high-contrast stimuli (50 or 100%) were presented only on the left or the right, with an unlimited choice window, and repeating trial conditions following incorrect choices ('repeat on incorrect'). Once mice achieved high accuracy and initiated movements rapidly – approximately 70 or 80% performance on non-repeat trials, and with reaction times nearly all <1 s – trials with no stimuli were introduced, again repeating on incorrect. Once subjects responded accurately on these trials (70 or 80% performance, at experimenter's discretion), lower contrast trials were introduced without repeat on incorrect. Finally, contrast comparison trials were introduced, starting with high vs. low contrast, then high vs. medium and medium

vs. low, then trials with equal contrast on both sides. The final proportion of trials presented was weighted towards easy trials (high contrast vs. zero, high vs. low, medium vs. zero, and no-stimulus trials) to encourage high overall reward rates and sustained motivation.

### Trial exclusion

Trials were excluded from analyses based on several criteria. For all experiments, error trials were excluded if they represented the second (or more) consecutive error on easy trials (see 'Repeat on incorrect' in previous section) as these errors likely reflect mouse disengagement. In addition, the first 5–10 trials of every session were excluded as these trials may include periods when the mouse is not settled into the task. For electrophysiological recording experiments, trials were excluded if they did not exhibit clear Left or Right wheel movements for Left and Right trials within 0.1–0.4 s from stimulus onset or if they exhibited twitch movements for NoGo trials within −0.05 to 0.5 s from stimulus onset.

## Widefield calcium imaging

### Mice and apparatus

Imaging was performed in transgenic mice expressing GCaMP6 in excitatory neurons (tetO-G6s × CaMK2a-tTA; VGlut1-cre × Ai95) or all neurons (Snap25-GCaMP6s). For all widefield analyses, data was averaged across these mouse genotypes since all groups showed qualitatively similar responses. Aberrant epileptiform activity has not been observed in these mouse lines (*Steinmetz et al., 2017*). Details of the imaging have been described before (*Jacobs et al., 2020*) and are summarized here. We imaged using a macroscope (Scimedia THT-FLSP) with sCMOS camera (PCO Edge 5.5) and dual-wavelength illumination (Cairn OptoLED). The macroscope used a 1.0× condenser lens (Leica 10450028) and 0.63× objective lens (Leica 10450027). Images were acquired from the PCO Edge with ~10 ms exposures and 2× two binning in rolling shutter mode. Images were acquired at 70 Hz, alternating between blue and violet illumination (35 Hz each). The light sources were 470 nm and 405 nm LEDs (Cairn OptoLED, P1110/002/000; P1105/405/LED, P1105/470/LED). Excitation light passed through excitation filters (blue: Semrock FF01-466/40-25; violet: Cairn DC/ET405/20x), and through a dichroic (425 nm; Chroma T425lpxr). Excitation light then went through 3 mm core optical fiber (Cairn P135/015/003) and reflected off another dichroic (495 nm; Semrock FF495-Di03-50x70) to the brain. Emitted light passed through the second dichroic and an emission filter (Edmunds 525/50-55 [86-963]) to the camera. Alternation was controlled with custom code on an Arduino Uno, and illumination was restricted to the 'global' phase of the rolling shutter exposures, that is, only the times when all pixels of a frame were being exposed together.

### Preprocessing

We de-noised the signal with singular value decomposition (*Peters et al., 2021*) and normalized the signal to the mean fluorescence at each pixel. The signal from the 405 nm illumination frames was used to correct for parts of the 470 nm signal that were due to changes in blood flow that obstruct the fluorescence signal (*Ma et al., 2016*) and the correction was performed with custom MATLAB code (http://www.github.com/cortex-lab/widefield; *Steinmetz and Peters, 2019*). We then low-pass filtered the signal at 8.5 Hz and applied a derivative filter to the fluorescence trace to approximate deconvolution of the calcium sensor's time course from the underlying neural activity. Fluorescence was extracted on a grid centered at bregma for each mouse and session, making it possible to average activity across sessions/mice. When computing stimulus-aligned averages of the fluorescence, pre-stimulus baseline activity was removed, removing the impact of long-term trends.

### ROI selection

Each mouse's cortical fluorescence map was aligned to the Allen Common Coordinate Framework (CCF) atlas (*Wang et al., 2020*). This alignment was performed manually by matching primary visual, secondary visual, and secondary motor areas to the corresponding hotspots of fluorescence following presentation of a contralateral stimulus. Single-pixel regions of interest (ROIs) for each cortical area were selected based on this atlas and manually adjusted to allow for inter-mouse differences. VISp was selected as the peak of the most posterior-medial activated site in the visual cortex in response to a contralateral stimulus. VISal was selected as the center of VISal according to the Allen

CCF. VISal was taken as an exemplary secondary visual cortical area because it was furthest from the part of VISp activated by our visual stimuli, ensuring minimal contamination of fluorescence between these two ROIs. The MOs' ROI was selected as the most anterior site activated by the contralateral stimulus, which was also within the CCF bounds for this region. MOp and SSp ROIs were selected within the cortical region active during wheel movements, positioned equidistant from the MOp-SSp border in the CCF.

## Decoding analysis

To decode task variables from neural activity at different cortical regions and time bins, we used a binary decoder, which measures how neural activity can predict one behavioral or sensory variable, while holding the others constant (*Steinmetz et al., 2019*), by generalizing choice probability analysis (*Britten et al., 1992*). We describe this method by explaining how it is used to predict Action Execution (Go vs. NoGo) for constant stimulus conditions. For each of the 16 possible stimulus conditions (contrast pairs), we compute a Mann–Whitney U statistic: the number of Go-NoGo trial pairs of this stimulus condition for which the Go trial had more activity than the NoGo trial. We then sum the 16 U statistics, sum the total number of trial pairs for each condition, and divide to obtain an area under receiver operating characteristic (auROC) value between 0 and 1. This auROC therefore quantifies how well a decoder could distinguish between Go and NoGo trials from neural activity, analogous to choice probability analysis but controlling for different stimulus conditions (*Britten et al., 1992*).

For Vision decoding, a similar strategy is used: to decode left contrast, we first divide trials into 12 groups corresponding to three possibilities for the animal's choice (Left, Right, NoGo) and four right stimulus contrasts. Within each group, we calculate a U statistic: the number of trials, for which one trial had non-zero contrast on the left and the other had zero contrast, and the non-zero contrast trial had more activity. We sum these U statistics across groups and divide by the total trial count to obtain an auROC value. For stimulus decoding on the right side, the same analysis is performed but swapping left for right contrast conditions. Contralateral and ipsilateral stimulus decoding is computed by combining the auROC values between hemispheres and hemifields. For Action Selection decoding, NoGo trials are excluded and a U statistic is computed between fluorescence for Left and Right choices, within each of the 16 stimulus conditions. The auROC value here gives an estimate for how well a decoder can discriminate Left from Right choices from neural activity, controlling for the stimulus contrast.

For the widefield dataset, decoding significance was measured for each coordinate and 25 ms time slice using a nested ANOVA across sessions and subjects (sessions nested within subjects), testing whether decoding performance was significantly different from 0.5. To compute Action decoding at movement-aligned time bins, NoGo trials were assigned a 'movement time' based on random sampling from the distribution of reaction times for correct Go trials.

## Optogenetic inactivation

While mice performed the task, we optogenetically inactivated several cortical areas through the skull using a blue laser. For these experiments, we utilized transgenic mice expressing ChR2 in parvalbumin-expressing inhibitory interneurons (Ai32 × *Pvalb*-Cre).

### 52-Coordinate inactivation experiment

Unilateral inactivation was achieved by mounting a fiber-optic cable (50 μm) with collimating and focusing lenses attached on a moving manipulator (Scientifica, Patch-Star). On every trial, custom code drove the manipulator to set the position of the fiber-optic cable to one of 52 different coordinates distributed across the cortex. Inactivation coordinates were defined stereotaxically from bregma. On ~75% of trials, the laser was switched on (473 nm, 1.5 mW, 40 Hz sine wave) to inactivate the cortical site. The laser dot was collimated and focused on the brain surface to ~100 μm radius, resulting in light power density of ~25 mW/mm$^2$ at the skull surface for the trial duration (peak power 1.5 mW)/(pi * (0.1 mm)$^2$), delivered in a sine wave such that average power is 50% of maximum, with additional power attenuation through the skull (*Guo et al., 2014*). Laser and non-laser trials, and the location of the cortical inactivation, were randomized. The duration of the laser

was from visual stimulus onset until a behavioral choice was made. The laser positioning was independent of laser power, so auditory noise from the manipulator did not predict inactivation.

For the majority of sessions, laser illumination was targeted uniformly at the 52 coordinates, thereby inactivating each coordinate a handful of times (~1.4% of trials). This discouraged any adaptation effects that may occur on more frequent inactivation paradigms. However, for nine sessions, the set of inactivation coordinates was restricted to coordinates within anterior MOs (2 mm AP, ±1.5 mm ML; 2 mm AP, ±0.5 mm ML; 3 mm AP, ±0.5 mm ML). For 14 sessions, the set of coordinates was restricted fall between VISp and SSp (−1 mm AP, ±2.5 mm ML; −2 mm AP, ±2.5 mm ML; −3 mm AP, ±2.5 mm ML). For these restricted coordinate sessions, the proportion of inactivated trials was reduced to 50%.

## Pulsed inactivation experiment

Unilateral pulsed inactivation was achieved using a pair of mirrors mounted on galvo motors to orient the laser (462 nm; collimated and focused to ~100 um dot size on cortical surface) to different points on the skull. We also introduced improved light isolation to ensure no light could reflect from the skull surface and be seen by the mouse. For ~66% of randomly interleaved trials, the laser was switched on for 25 ms (DC) at random times relative to stimulus onset (−300 to +300 ms drawn from a uniform distribution). Inactivation was targeted at visual areas (VIS; −4 mm AP, ±2 mm ML), secondary motor area (MOs; +2 mm AP, ±0.5 mm ML), and primary motor area (MOp; −0.5 mm AP, ±1 mm ML). The VIS coordinate was chosen stereotaxically as the medial part of VISp, corresponding to the part of VISp with retinotopy for the stimulus location. The MOs' coordinate was chosen in anterior MOs, corresponding to the region that exhibited significant behavioral effects from the 52-coordinate inactivation experiment. The MOp coordinate was selected as the most posterior part of MOp from the CCF atlas, positioned as far as possible from MOs to reduce the possibility that MOp inactivation also silences MOs' activity.

## Mixed-power inactivation experiment

Unilateral inactivation was achieved using the same laser/galvo hardware as in the pulsed inactivation experiment. The mirrors oriented the laser at visual and secondary motor areas. For ~70% of randomly interleaved trials, the laser was switched on from stimulus onset for 1.5 s fixed duration. Laser power was chosen randomly between 1.5, 2.9, and 4.2 mW. For subsequent analyses, trials using different laser powers were pooled together.

## Electrophysiological recordings

The Neuropixels electrophysiological dataset was described previously (*Steinmetz et al., 2019*). Methods relating to electrophysiological recordings have been detailed in this work and are reproduced here.

### Hardware

Recordings were made using Neuropixels electrode arrays (*Jun et al., 2017*). Probes were mounted to a custom 3D-printed PLA piece and affixed to a steel rod held by a micromanipulator (uMP-4, Sensapex Inc). To allow later track localization, prior to insertion probes were coated with a solution of DiI (ThermoFisher Vybrant V22888 or V22885) by holding 2 μL in a droplet on the end of a micropipette and touching the droplet to the probe shank, letting it dry, and repeating until the droplet was gone, after which the probe appeared pink.

### Procedure

On the day of recording or within 2 days before, mice were briefly anesthetized with isoflurane while one or more craniotomies were made, either with a dental drill or a biopsy punch. The craniotomies for VISp were targeted in some cases using measured retinotopic maps in the same mice, and in other cases to the same position stereotaxically (−4 mm AP, 1.7 mm ML, left hemisphere). The craniotomies for MOs were targeted stereotaxically (+2 mm AP, 0.5 mm ML, left hemisphere) to match the coordinates with strong inactivation effects. After at least 3 hr of recovery, mice were head-fixed in the setup. Probes had a soldered connection to short external reference to ground; the ground connection at the headstage was subsequently connected to an Ag/AgCl wire positioned on the

skull. The craniotomies as well as the wire were covered with saline-based agar. The agar was covered with silicone oil to prevent drying. In some experiments, a saline bath was used rather than agar. Probes were advanced through the agar and through the dura, then lowered to their final position at ~10 µm/s. Electrodes were allowed to settle for ~15 min before starting recording. Recordings were made in external reference mode with LFP gain = 250 and AP gain = 500. Recordings were repeated on multiple subsequent days. All recordings were made in the left hemisphere.

### Preprocessing

The data were automatically spike sorted with Kilosort (*Pachitariu et al., 2016*; http://www.github.com/cortex-lab/Kilosort) and then manually curated with the 'phy' gui (http://www.github.com/kwik-team/phy). Extracellular voltage traces were preprocessed using common-average referencing: subtracting each channel's median to remove baseline offsets, then subtracting the median across all channels at each time point to remove artifacts. During manual curation, each set of events ('unit') detected by a particular template was inspected, and if the spikes assigned to the unit resembled noise (zero or near-zero amplitude; non-physiological waveform shape or pattern of activity across channels), the unit was discarded. Units containing low-amplitude spikes, spikes with inconsistent waveform shapes, and/or refractory period contamination were labeled as 'multi-unit activity' and not included for further analysis. Finally, each unit was compared to similar, spatially neighboring units to determine whether they should be merged, based on spike waveform similarity, drift patterns, or cross-correlogram features. For calculating event-triggered averages and decoding performance, spike counts were binned with a 1 ms window and then smoothed with a 25 ms causal Gaussian filter.

### Decoding analysis

Vision, Action Execution, and Action Selection were decoded from electrophysiological recordings using the same binary decoder analysis outlined above for widefield decoding, with one exception: since single-neuron data could not be averaged across sessions, statistical significance could not be assessed by comparing decoding performance across sessions. Instead, decoding performance as measured by auROC was compared to a shuffled null distribution of auROC values obtained by shuffling the trial labels 2000 times (e.g., shuffling Go vs. NoGo trial labels for Action Execution decoding).

### Kernel analysis

The cross-validated variance of each neuron's firing rate explained by Vision, Action, or Choice was determined using a reduced-rank kernel regression method detailed previously (*Steinmetz et al., 2019*). In brief, a predictor matrix was formed using the times and identities of stimuli and actions. Reduced-rank regression between the predictor matrices and the binned spike counts was performed across all recordings simultaneously. The variance explained by the regression was assessed with cross-validation. Vision/Action/Choice kernels were deemed significant if their cross-validated variance explained exceeded 2%. The false-positive rate for this threshold was found with a shuffle test to be 0.33%.

## Psychometric model

We modeled probabilistic choice behavior using a multinomial logistic regression described previously (*Burgess et al., 2017*), but with the addition of a hierarchical Bayesian framework to account for inter-subject and inter-session variability in the model parameters.

For each trial $i$, the probability ratio of Left vs. NoGo and Right vs. NoGo choices is set by the exponential function of two decision variables $Z_L^{(i)}$ and $Z_R^{(i)}$:

$$P^{(i)}(Left)/P^{(i)}(NoGo) = exp\left(Z_L^{(i)}\right)$$

$$P^{(i)}(Right)/P^{(i)}(NoGo) = exp\left(Z_R^{(i)}\right)$$

$$P^{(i)}(NoGo) = 1 - P^{(i)}(Left) - P^{(i)}(Right)$$

Choices $y^{(i)} \in (Left, Right, NoGo)$ are drawn from a categorical probability distribution with these parameters:

$$y^{(i)}$$

To capture the behavioral dependence on choice bias and stimulus sensitivity, the decision variables are modeled as a saturating nonlinear transformation of stimulus contrast. We denote the session number in which trial $i$ occurred as $d[i]$, and the subject (mouse) number that performed the session as $m[d]$. The decision variables depended on parameters that varied between sessions (and thus also between subjects) according to the following formulae:

$$Z_L^{(i)} = b_L^{d[i]} + s_L^{d[i]} \cdot \left( c_L^{(i)} \right)^{n^{d[i]}}$$

$$Z_R^{(i)} = b_R^{d[i]} + s_R^{d[i]} \cdot \left( c_R^{(i)} \right)^{n^{d[i]}}$$

Here, $b_L^{d[i]}$ and $b_R^{d[i]}$ are bias parameters, which capture stimulus-independent choice behavior in session $d$, while $s_L^{d[i]}$ and $s_R^{d[i]}$ are session-dependent sensitivity parameters scaling the visual input on the left and right sides. The visual input consists of the contrast presented on the left ($c_L^{(i)}$) and right ($c_R^{(i)}$), raised to a session-dependent exponent parameter $0 \le n^{d[i]} \le 1$ to allow for a saturating nonlinear contrast transformation.

To capture how these parameters vary across sessions and subjects, we expanded the model to incorporate a hierarchical prior on sessions and subjects. Let $\theta_d$ be a five-element vector containing the five session-specific parameters stated above, $\theta_d = \left[ b_L^{(d)}, b_R^{(d)}, s_L^{(d)}, s_R^{(d)}, n^{(d)} \right]$. We model each session's parameter vector $\theta_d$ as drawn from a multivariate Gaussian distribution whose mean $\underline{\theta}_{m[d]}$ depends on the subject, with a common covariance matrix $\Sigma$,

$$\theta_d$$

The covariance matrix $\Sigma$ is given the following prior. It is first converted to a correlation matrix, which is given a $LKJ(2)$ prior (*Lewandowski et al., 2009*) to penalize large positive or negative parameter correlations across sessions. The standard deviation terms for each parameter are given a $HalfCauchy(0,1)$ prior to penalize large variability in each parameter across sessions.

The subject-level mean vector $\underline{\theta}_m$ is drawn from a Gaussian grand-average mean $\underline{\theta}^*$, with a covariance matrix $\Sigma^*$, which quantifies covariation in the parameters across subjects,

$$\underline{\theta}_m \sim N(\underline{\theta}^*, \Sigma^*)$$

The covariance matrix $\Sigma^*$ is given the same prior as $\Sigma$. Finally, the grand-average parameters are given a weakly informative Gaussian hyperprior with mean $[0,0,5,5,0.5]$, variances $[2,2,2,2,0.25]^2$, and covariances all zero.

The full joint posterior distribution of all parameters was numerically estimated with Hamiltonian Monte Carlo (No-U-Turn) sampling using the Stan language (*Carpenter et al., 2017*). Sampling was performed in four chains, each with 500 warmup iterations and 500 sampling iterations. The samples were checked manually to ensure convergence within and between chains. The posterior prediction/fit from the model is constructed by computing the model prediction from each of the posterior distribution samples, and then computing the mean and 95% credible intervals on the prediction across samples. All model predictions shown in the figures use the grand-average parameter posterior $\underline{\theta}^*$, unless specified otherwise.

## Neurometric model

The neurometric model predicts the animal's choice from the activity of ROIs in left VISp, right VISp, left MOs, and right MOs. Like the psychometric model, it has two decision variables, which are determined on each trial from a weighted sum of activity in the four cortical areas:

$$Z_L^{(i)} = \alpha_L^{d[i]} + f^{(i)}.w_L^{d[i]}$$

$$Z_R^{(i)} = \alpha_R^{d[i]} + f^{(i)}.w_R^{d[i]}$$

The decision variable are thus a sum of offsets $\alpha_L^{d[i]}$ and $\alpha_R^{d[i]}$, and the inner product between two $4 \times 1$ session-dependent weight vectors $w_L^{d[i]}$ and $w_R^{d[i]}$, and a vector $f^{(i)}$ containing the estimated population firing rate in the four regions on trial $i$. The weights and offset parameters are given a hierarchical prior allowing for variation between sessions and subjects similar to in the psychometric model:

$$\underline{\alpha_L}^*, \underline{\alpha_R}^* \sim N\left(0, 4^2\right)$$

$$\underline{w_L}^*, \underline{w_R}^* \sim N\left(0, I \times 4^2\right)$$

The estimated population rate $f^{(i)}$ is derived from widefield calcium imaging at the four ROIs (see Widefield methods for detail). In subsequent analyses, $f^{(i)}$ will be set to zero to generate behavioral predictions for cortical inactivation. However, widefield dF/F data has no meaningful baseline due to background fluorescence. Therefore, prior to fitting the model, widefield dF/F data is calibrated to electrophysiologically measured firing rates. This calibration was achieved by recording extracellular spiking activity in VISp and MOs using Neuropixels probes in separate sessions and computing trial-averaged firing rates for each of the contrast conditions over a time window (*Figure 4—figure supplement 1*; VISp: 75–125 ms, MOs: 125–175 ms). Calcium fluorescence was also averaged over the same windows but 30 ms later to allow for slower GCaMP6 kinetics. The transformation from widefield fluorescence to firing rate was computed by simple linear regression over the 16 contrast conditions. This linear transformation was then applied to the fluorescence value for each individual trial, thereby providing a population firing rate estimate for the four cortical regions on every trial. To improve fit stability, parameter symmetry was enforced between the left and right hemispheres (e. g., the weight of left VISp onto $Z_L$ was the same as right VISp onto $Z_R$).

To generate predictions from the neurometric model for the effects of optogenetic inactivation, the model was modified in two ways. Firstly, since neural activity in VISp and MOs was not measured during optogenetic inactivation sessions, the trial-by-trial activity in $f^{(i)}$ was replaced with the trial-averaged firing rate measured electrophysiologically for each contrast condition. Secondly, since the overall tendency to NoGo differed idiosyncratically between widefield imaging and optogenetic inactivation sessions, the model offset parameters ($\alpha_L$ and $\alpha_R$) were re-fit to the non-laser trials contained within optogenetic inactivation sessions (Mixed-power inactivation experiment).

To simulate the effect of optogenetic inactivation of a single cortical area, one element of the firing rate vector $f$ was set to zero. This effect propagates forward through the model based on the fixed weights estimated from the widefield sessions, thereby affecting the decision variables and the probability associated with each choice. Importantly, the behavioral prediction obtained from the model when simulating inactivation did not depend on any empirical data involving actual optogenetic inactivation. In this sense, the neurometric model predicts behavior in a new dataset on which it was not fit.

## Statistical tests

For the 52-coordinate inactivation experiment (*Figure 3a*), statistical significance of the inactivation effect was assessed using a permutation test. The test statistic used was the difference in the proportion of a specific choice type, between laser and non-laser off trials (on trials with equal left and right contrast) for each of the inactivated coordinates, averaged across sessions. The null distribution

of the test statistic was computed by repeated shuffling of laser and non-laser trial identities within each session. All other statistical tests are specified in the main text.

## Acknowledgements

We thank Michael Krumin for assistance in the design and construction of the galvo-controlled laser stimulation setup; Charu Reddy, Miles Wells, Laura Funnell, and Hamish Forrest for help with mouse husbandry and training; Pip Coen, Kevin Miller, Hamish Forrest, and Andy Peters for feedback on earlier forms of the manuscript.

## Additional information

### Funding

| Funder | Grant reference number | Author |
| --- | --- | --- |
| Wellcome Trust | 095668 | Matteo Carandini<br>Kenneth D Harris |
| Wellcome Trust | 095669 | Matteo Carandini |
| Human FrontiersScience Program | LT001071 | Nicholas A Steinmetz |
| Horizon 2020 | 656528 | Nicholas A Steinmetz |
| Engineering and Physical Sciences Research Council | CoMPLEX PhD studentship | Peter Zatka-Haas |
| European Research Council | 694401 | Kenneth D Harris |

The funders had no role in study design, data collection and interpretation, or the decision to submit the work for publication.

### Author contributions

Peter Zatka-Haas, Nicholas A Steinmetz, Conceptualization, Data curation, Software, Formal analysis, Validation, Investigation, Visualization, Methodology, Writing - original draft, Writing - review and editing; Matteo Carandini, Kenneth D Harris, Conceptualization, Supervision, Funding acquisition, Visualization, Methodology, Writing - original draft, Project administration, Writing - review and editing

### Author ORCIDs

Peter Zatka-Haas (iD) https://orcid.org/0000-0001-9054-2599

Nicholas A Steinmetz (iD) https://orcid.org/0000-0001-7029-2908

Matteo Carandini (iD) https://orcid.org/0000-0003-4880-7682

Kenneth D Harris (iD) https://orcid.org/0000-0002-5930-6456

### Ethics

Animal experimentation: All experimental procedures were conducted at UCL according to the UK Animals Scientific Procedures Act (1986) and under personal and project licenses granted by the Home Office following appropriate ethics review.

### Decision letter and Author response

Decision letter https://doi.org/10.7554/eLife.63163.sa1

Author response https://doi.org/10.7554/eLife.63163.sa2

# Additional files

## Supplementary files
• Supplementary file 1. Mouse genotype and session count for each figure. Table lists the mice used in this study, sex/genotype information, and the number of sessions used for the data in each figure panel.

• Transparent reporting form

## Data availability
The behavioural and neural datasets generated and analyzed in this study are available as downloadable files. Neuropixels electrophysiological data is available at https://figshare.com/articles/steinmetz/9598406. Widefield calcium imaging data, optogenetic inactivation data, model fits and associated behavioural data and code are available at https://figshare.com/articles/dataset/Zatka-Haas_et_al_2020_dataset/13008038.

The following previously published dataset was used:

| Author(s) | Year | Dataset title | Dataset URL | Database and Identifier |
|---|---|---|---|---|
| Steinmetz N, Zatka-Haas P, Carandini M, Harris K | 2019 | Main dataset from Steinmetz et al. 2019 | https://figshare.com/articles/dataset/Dataset_from_Steinmetz_et_al_2019/9598406/2 | figshare, 10.6084/m9.figshare.9727895 |

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
