## [Decision Letter]

**Acceptance summary:**

Perceptual decisions are often associated with widespread changes in neural activity, encompassing sensory signals, choice signals and signals relating to motor behavior. However, what is the causal relevance of these neuronal changes? Here, Zatka-Haas disentangle different components of a behavioral task, namely action selection, action initiation and sensory encoding, and characterize their unique and distributed correlates using large-scale imaging and Neuropixel recordings. Subsequently, large-scale and targeted optogenetic inactivation was used to determine which cortical regions have a direct causal relevance for behavior. Specific regions in sensory and frontal cortex made direct causal contributions to behavior in a way that can be predicted precisely from their sensory correlates. By contrast, regions with action initiation correlates were not directly causally relevant for the task, despite the fact that these regions distinguished between hits and misses.

**Decision letter after peer review:**

[Editors’ note: the authors submitted for reconsideration following the decision after peer review. What follows is the decision letter after the first round of review.]

Thank you for submitting your work entitled "A perceptual decision requires sensory but not action coding in mouse cortex" for consideration by *eLife*. Your article has been reviewed by 3 external peer reviewers, and the evaluation has been overseen by a Reviewing Editor and a Senior Editor. The following individual involved in review of your submission has agreed to reveal their identity: Karel Svoboda (Reviewer #2).

Our decision has been reached after careful consultation between the reviewers. Based on these discussions and the individual reviews below, we regret to inform you that your work will not be further considered for publication in *eLife*. Reviewers made several positive comments about the ambitious nature of the study as well as the quality of the dataset. However, major concerns were raised about the interpretation of the neurometric model, the interpretation of the task correlates, and the general conceptual advance made by the study.

*Reviewer #2:*

The authors used widefield imaging, Neuropixels recording, optogenetic inactivation, and modeling to study the coding and causal roles of dorsal cortex in a two-alternative unforce-choice task. Consistent with the ephys data reported previously by the same lab (Steinmetz et al. 2019), widefield imaging revealed distinct patterns of cortical activity related to vision, action, and choice. Interestingly, whereas the vision correlates match their causal roles both spatially and temporally, the widespread action correlates appear to be non-causal. Based on these observations, the authors constructed a neurometric model that linearly combines population activity in visual and frontal cortex. This simple model captured the animal's behavior and predicted the difference in the inactivation effects of the two regions.

Overall, this is an interesting study that puts together a variety of experimental and computational techniques to build quantitative links among different measurements of the same task. However, the data are not completely self-consistent, and the interpretations of the results are not fully satisfying. In addition, the key findings of this study (pre-movement cortical choice coding is weak; cortical action coding is non-causal) are more likely to be task-specific rather than generalizable (as already implied by the authors).

1. Although it is nice to use multiple methods, widefield imaging and ephys record very different things. This comparson should be explored better. Where do the widefield signals come from? How does it compare to the depth distribution sampled from Neuropixels recordings? Is there any laminar signature in the ephys data? For example, in Figure 2n, the SSp trace does not rise until -50 ms, but in the imaging data (Figure 2g), significant action signal already appears in SSp at -100 ms. Could this be explained by a sampling bias towards superficial layers in widefield imaging?

2. Encoding versus decoding:

(1) In the encoding model, 20.1% of SSp neurons had significant pre-movement action encoding (Line 141), which was stronger than MOp (15.3%), whereas in Figure 2n, the action decoding of SSp is clearly weaker than MOp, with a proportion of significant pre-movement decoding far less than 20%. How csn this discrepancy be explained?

(2) Line 56 "correlates of action.… strongest in primary motor and somatosensory cortex" and Line 390 "What then might be the function of the strong MOp/SSp activity observed prior to action execution?" Given the above discrepancy, how to define the strength of "action correlates"? By encoding or decoding? For example, based on decoding (Figure 2n), one would conclude that the strongest action correlates prior to movement is in MOp/MOs, or even VISal, but not in SSp.

3. The main conclusion of this study, as manifested in the title, is that the widespread cortical action coding is not causal. But some concerns in data analysis and result interpretation may weaken this conclusion.

(1) The conclusion is based on the negative results in Figure 3d-f, which averaged over equal non-zero contrast trials. Are these the best stimulus conditions to look at? It is reasonable to use these conditions in Figure 3a-c since the balanced left and right choices on control trials may maximize any Δ Contraversive effect of inactivation. However, following the same logic, it is more likely to observe significant ΔNoGo, if any, when the Go and NoGo choices are balanced, i.e., on trials with zero contrast on both sides (~50% NoGo in Figure 1f). In other words, using non-zero contrast trials, Figure 3d-f could have substantially underestimated the inactivation effect on Action.

(2) Also related to point (1), Figure 4d suggests that VIS or MOs inactivation does increase the NoGo choices for at least unilateral stimulus conditions. Although these two areas were assumed to be only sensory-related in the context of Figure 4, they also encode action (Figure 2). In particular, the pre-movement action coding in MOs is comparable with MOp (Figure 2n). Therefore, how to reconcile the observation in Figure 4d with the conclusion that cortical action coding is not causal?

(3) One argument weakening the conclusion is that the localized unilateral inactivation could be too weak to perturb distributed action coding. The authors point this out (Line 381), but it is hard to tell whether the current interpretation is "most parsimonious" (Line 385). Without bilateral and multi-regional inactivation data, the overall conclusion of this study should be more conservative. Instead of saying "do not play a causal role", descriptions like "not locally causal" (Line 415) may be more appropriate.

4. The neurometric model seems oversimplified. It uses one scalar variable to represent the population activity of each area, with an underlying assumption that all the areas are homogeneous. But this is not the case for MOs, where contra- and ipsi-lateral visual coding are mixed (Figure 2j and m). This may explain the strange result that the weights of MOs are additive (Figure 4b), which should be discussed. Is it possible to fit a similar neurometric model using the ephys data that at least takes this level of heterogeneity into account?

5. Many different protocols have been used throughout the paper, but the motivations are not always clear, making the results sometimes difficult to compare. For example:

(1) The open-loop period was used in imaging and ephys sessions, but not in the inactivation experiments. Did the animals behave differently with and without the open-loop period?

(2) In the pulse inactivation experiment, 15 mW 25 ms photostimuli were used, but in Figure 3 Sup. 3c and f, the simultaneous ephys recordings were done with 4 mW, 10 ms. It is unclear how different these two protocols could be in terms of the long-lasting suppression effect, which is important for interpreting Figure 3h and i. Moreover, the rebound after 100 ms seems to be strong in Figure 3 Sup. 3c and f. Could it be even stronger in 15 mW 25 ms case? Rebound activity can greatly confound interpretation of inactivation experiments.

(3) In Figure 4, multiple laser powers were used, but the data were pooled together to compare with the neurometric model. Is this valid given that the effect of MOs inactivation depends on the laser power Figure 3 Sup.1f? This also raises a concern related to major point 3(3): what if higher powers are used in Figure 3d-f? It seems unfair to include 2.9 and 4.25 mW in Figure 4 while drawing conclusion from Figure 3d-f using only 1.5 mW.

*Reviewer #3:*

This is a very ambitious study using an impressive set of cutting-edge techniques. The authors' goal is to address the timely and interesting question "where in the cortex does neural activity code for sensation, choice, and action?" They present data from wide-field imaging, single-unit recordings, and targeted optogenetic inactivation, along with a model. They ultimately find neural representations of sensation and motor action, but very little encoding of choice, across the cortical mantle, and this is reinforced by the results of cortical inactivation. Overall, the lack of truly novel results and the incomplete nature of the findings reduce the potential impact.

The study falls short for two reasons. First, this may simply be the wrong task in which to examine choice representation in the cortex. As the authors nicely show, mice begin turning the wheel immediately after the stimulus presentation, and no delay is enforced. To deeply examine the question posed by the authors, the task would need to have stimulus, delay, and response periods separated to allow the experimenter to observe and perturb uncontaminated patterns of neural activity.

Second, the authors may not be using the right approach to detect key populations of cortical neurons. They suggest that the sparseness of neurons that encode choice indicates that the cortex simply does not generally function in this aspect of the behavior. They further point to the idea that subcortical areas such as the colliculus and the striatum have been implicated in visual detection tasks. However, the authors do not include data showing that choice is in fact encoded by neurons in those regions. Recent studies (Lee at al, Cell Reports 2020; Tang et al. Neuron 2020; Puscian et al. Cell Reports 2020, etc) have in fact shown that neurons in primary visual cortex robustly encode choice in visual detection tasks. However, several of these papers highlighted that this encoding is a feature of neural populations with projection targets specific to the type of task. Thus, Neuropixels recordings of unidentified neurons may give the mistaken impression that choice encoding is extremely sparse or not present in the cortex, but targeted recordings or imaging of specific populations may reveal that such encoding is robust. Such a finding would be entirely in keeping with the growing realization in the field that each cortical area may represent physically intermingled but functionally separable components of large-scale circuits.

*Reviewer #4:*

Understanding the relationships between neural activity and function continues to be a major undertaking. This study uses widefield calcium imaging and Neuropixel recording data to interpret causal impacts of optogenetic suppression sampled across dorsal cortex. The authors present the novel findings that (1) effects of optogenetic suppression are specifically correlated with the magnitude of sensory encoding within each region and (2) frontal and occipital cortices make different contributions to choice formation (substrative vs additive).

The experiments and analyses are performed at high quality. The insights are novel, interesting, and relevant to a general neuroscience audience. The points below attempt to clarify some of the major findings.

– I encourage the authors to carefully acknowledge what is novel in this study as opposed to published work from recent studies (eg, Steinmetz et al., 2019). For example, the presentation of single unit choice probability in the Results and Discussion do not appropriately acknowledge which analyses and assessments are novel.

– I am concerned with the central of the logic of the study as presented. If we consider the framework that decision-making occurs before action/response, then of course we would not expect action coding to impact 'a perceptual decision'. And yet, given the title, the authors appear to consider this a core finding of their study.

– (Related to above) I would expect action coding to causally impact motor performance. Indeed, the authors report that MOp suppression reduced peak wheel velocity by 20%. It seems odd that this important finding is minimized in the Results and Discussion. This finding argues strongly that the action signals in MOp do not solely reflect a corollary discharge. Did suppression in other 'action coding' regions also alter motor performance (wheel velocity, reaction time, other measures)?

– What is the reason for the additional optogenetic suppression experiments presented in in Figure 4 (how are they different from the experiments presented in Figure 3)? Are the lack of effects on NoGo probability in Figure 3D inconsistent with the increased Misses in Figure 4D? Is the lack of increased incorrect responses in MOs in Figure 4D inconsistent with the increased rightward choices in MOs in Figure 3A?

[Editors’ note: further revisions were suggested prior to acceptance, as described below.]

Thank you for choosing to send your work entitled "A perceptual decision requires sensory but not action coding in mouse cortex" for consideration at *eLife*. Your article has been considered by a Senior Editor and a Reviewing editor, and we are prepared to consider a revised submission.

Please take note of the following points when preparing your revised submission:

Reviewers made several positive comments about the ambitious nature of the study as well as the quality of the dataset, which combines many different techniques in a technically impressive manner. Reviewers generally commented positively on the identification of causally related task correlates related to sensory perception, which is based on a combination of optogenetics, recordings and modelling. However, they still had major concerns about the interpretation and analysis of the choice and action correlates, including about issues covered in the action plan that should be considered further, as follows:

1) Explain conceptual advance

The authors should better explain the novelty of their findings and the relationship to previous work (in particular Steinmetz et al. 2019). For example it needs to be clarified which aspects of findings using the same dataset are novel, e.g. the presentation of single unit choice probability.

2) Improve overall conceptual interpretation of task correlates and lack of choice correlates

(i) The study should explain better how to understand the concept of choice in light of their task and findings. The authors should discuss why this is an appropriate task to study choice representation in the cortex, and how the interpretations of the authors might be constrained by not having separate stimulus, delay and response periods. The authors should improve their explanation of the difference between choice vs. no/go correlates.

(ii) The authors should clarify the central of the logic of the study. If we consider the framework that decision-making occurs before action/response, then one would not expect action coding to impact 'a perceptual decision'. And yet, given the title, the authors appear to consider this a core finding of their study.

(iii) One argument weakening the conclusion about the finding that widespread action coding is not causal, is that the localized unilateral inactivation could be too weak to perturb distributed action coding. The authors point this out (Line 381), but it is hard to tell whether the current interpretation is "most parsimonious" (Line 385). Without bilateral and multi-regional inactivation data, the overall conclusion of this study should be more conservative. Instead of saying "do not play a causal role", descriptions like "not locally causal" (Line 415) may be more appropriate.

(iv) The authors should address the apparent discrepancy with studies (e.g. Lee at al, Cell Reports 2020; Tang et al. Neuron 2020; Puscian et al. Cell Reports 2020, etc) that show that neurons in primary visual cortex robustly encode choice in visual detection tasks.

(v) The authors should discuss the possibility that targeted recordings or imaging of specific populations according to projection patterns may reveal that choice encoding is robust, following the idea that there could be functionally separable components of large-scale circuits, and that choice encoding might a feature of neural populations with projection targets specific to the type of task.

3) Interpretation and analysis of action correlates

The main conclusion of this study, as manifested in the title, is that the widespread cortical action coding is not causal. The authors should address several concerns in data analysis and result interpretation that may weaken this conclusion.

(i) The conclusion is based on the negative results in Figure 3d-f, which averaged over equal non-zero contrast trials. Are these the best stimulus conditions to look at? It is reasonable to use these conditions in Figure 3a-c since the balanced left and right choices on control trials may maximize any Δ Contraversive effect of inactivation. However, following the same logic, it is more likely to observe significant ΔNoGo, if any, when the Go and NoGo choices are balanced, i.e., on trials with zero contrast on both sides (~50% NoGo in Figure 1f). In other words, using non-zero contrast trials, Figure 3d-f could have substantially underestimated the inactivation effect on Action.

(ii) Also related to point (3-i), Figure 4d suggests that VIS or MOs inactivation does increase the NoGo choices for at least unilateral stimulus conditions. Although these two areas were assumed to be only sensory-related in the context of Figure 4, they also encode action (Figure 2). In particular, the pre-movement action coding in MOs is comparable with MOp (Figure 2n). Therefore, how to reconcile the observation in Figure 4d with the conclusion that cortical action coding is not causal?

(iii) Related to the question whether action coding causally impact motor performance: Indeed, the authors report that MOp suppression reduced peak wheel velocity by 20%. The authors should discuss this finding more prominently in the Results and Discussion. This finding argue that the action signals in MOp do not solely reflect a corollary discharge. Did suppression in other 'action coding' regions also alter motor performance (wheel velocity, reaction time, other measures)?

4) Comparison between encoding vs. decoding results

(i) In the encoding model, 20.1% of SSp neurons had significant pre-movement action encoding (Line 141), which was stronger than MOp (15.3%), whereas in Figure 2n, the action decoding of SSp is clearly weaker than MOp, with a proportion of significant pre-movement decoding far less than 20%. How can this discrepancy be explained?

(ii) Line 56 "correlates of action.… strongest in primary motor and somatosensory cortex" and Line 390 "What then might be the function of the strong MOp/SSp activity observed prior to action execution?" Given the above discrepancy, how to define the strength of "action correlates"? By encoding or decoding? For example, based on decoding (Figure 2n), one would conclude that the strongest action correlates prior to movement is in MOp/MOs, or even VISal, but not in SSp.

5) Comparison between modalities

The comparison between different methods, in particular widefield imaging and ephys recordings, should be explored better. Where do the widefield signals come from? How does it compare to the depth distribution sampled from Neuropixels recordings? Is there any laminar signature in the ephys data? For example, in Figure 2n, the SSp trace does not rise until -50 ms, but in the imaging data (Figure 2g), significant action signal already appears in SSp at -100 ms. Could this be explained by a sampling bias towards superficial layers in widefield imaging?

6) Use of CCCP

CCCP, as a measure of choice probability, appears to be misleading when used to describe analyses to quantify sensory and action coding. One possibility is to introduce the general method of Combined Conditions Probability, and apply this analysis to Stimulus, Choice, or Action. (sCCP, cCCP, aCCP).

7) Comparison between protocols

Many different protocols have been used throughout the paper, but the motivations are not always clear, making the results sometimes difficult to compare. For example:

(i) The open-loop period was used in imaging and ephys sessions, but not in the inactivation experiments. Did the animals behave differently with and without the open-loop period?

(ii) In the pulse inactivation experiment, 15 mW 25 ms photostimuli were used, but in Figure 3 Sup. 3c and f, the simultaneous ephys recordings were done with 4 mW, 10 ms. It is unclear how different these two protocols could be in terms of the long-lasting suppression effect, which is important for interpreting Figure 3h and i. Moreover, the rebound after 100 ms seems to be strong in Figure 3 Sup. 3c and f. Could it be even stronger in 15 mW 25 ms case? Rebound activity can greatly confound interpretation of inactivation experiments.

(iii) In Figure 4, multiple laser powers were used, but the data were pooled together to compare with the neurometric model. Is this valid given that the effect of MOs inactivation depends on the laser power Figure 3 Sup.1f? This also raises a concern related to major point 3(3): what if higher powers are used in Figure 3d-f? It seems unfair to include 2.9 and 4.25 mW in Figure 4 while drawing conclusion from Figure 3d-f using only 1.5 mW.

(iv) What is the reason for the additional optogenetic suppression experiments presented in in Figure 4 (how are they different from the experiments presented in Figure 3)? Are the lack of effects on NoGo probability in Figure 3D inconsistent with the increased Misses in Figure 4D? Is the lack of increased incorrect responses in MOs in Figure 4D inconsistent with the increased rightward choices in MOs in Figure 3A?

8) Neurometric model related to sensory correlates:

The authors should discuss why the relatively simple neurometric model is appropriate. It uses one scalar variable to represent the population activity of each area, with an underlying assumption that all the areas are homogeneous. But this is not the case for MOs, where contra- and ipsi-lateral visual coding are mixed (Figure 2j and m). This may explain the strange result that the weights of MOs are additive (Figure 4b), which should be discussed. Is it possible to fit a similar neurometric model using the ephys data that at least takes this level of heterogeneity into account? The authors should address why a model with additive weights for MO and visual cortex is appropriate given that one might expect MO activity to depend on activity in the visual cortex.

---

## [Author Response]

[Editors’ note: The authors appealed the original decision. What follows is the authors’ response to the first round of review.]

Reviewer #2:The authors used widefield imaging, Neuropixels recording, optogenetic inactivation, and modeling to study the coding and causal roles of dorsal cortex in a two-alternative unforce-choice task. Consistent with the ephys data reported previously by the same lab (Steinmetz et al. 2019), widefield imaging revealed distinct patterns of cortical activity related to vision, action, and choice. Interestingly, whereas the vision correlates match their causal roles both spatially and temporally, the widespread action correlates appear to be non-causal. Based on these observations, the authors constructed a neurometric model that linearly combines population activity in visual and frontal cortex. This simple model captured the animal's behavior and predicted the difference in the inactivation effects of the two regions.Overall, this is an interesting study that puts together a variety of experimental and computational techniques to build quantitative links among different measurements of the same task. However, the data are not completely self-consistent, and the interpretations of the results are not fully satisfying. In addition, the key findings of this study (pre-movement cortical choice coding is weak; cortical action coding is non-causal) are more likely to be task-specific rather than generalizable (as already implied by the authors).Reviewer #3:This is a very ambitious study using an impressive set of cutting-edge techniques. The authors' goal is to address the timely and interesting question "where in the cortex does neural activity code for sensation, choice, and action?" They present data from wide-field imaging, single-unit recordings, and targeted optogenetic inactivation, along with a model. They ultimately find neural representations of sensation and motor action, but very little encoding of choice, across the cortical mantle, and this is reinforced by the results of cortical inactivation. Overall, the lack of truly novel results and the incomplete nature of the findings reduce the potential impact.The study falls short for two reasons. First, this may simply be the wrong task in which to examine choice representation in the cortex. As the authors nicely show, mice begin turning the wheel immediately after the stimulus presentation, and no delay is enforced. To deeply examine the question posed by the authors, the task would need to have stimulus, delay, and response periods separated to allow the experimenter to observe and perturb uncontaminated patterns of neural activity.Second, the authors may not be using the right approach to detect key populations of cortical neurons. They suggest that the sparseness of neurons that encode choice indicates that the cortex simply does not generally function in this aspect of the behavior. They further point to the idea that subcortical areas such as the colliculus and the striatum have been implicated in visual detection tasks. However, the authors do not include data showing that choice is in fact encoded by neurons in those regions. Recent studies (Lee at al, Cell Reports 2020; Tang et al. Neuron 2020; Puscian et al. Cell Reports 2020, etc) have in fact shown that neurons in primary visual cortex robustly encode choice in visual detection tasks. However, several of these papers highlighted that this encoding is a feature of neural populations with projection targets specific to the type of task. Thus, Neuropixels recordings of unidentified neurons may give the mistaken impression that choice encoding is extremely sparse or not present in the cortex, but targeted recordings or imaging of specific populations may reveal that such encoding is robust. Such a finding would be entirely in keeping with the growing realization in the field that each cortical area may represent physically intermingled but functionally separable components of large-scale circuits.Reviewer #4:Understanding the relationships between neural activity and function continues to be a major undertaking. This study uses widefield calcium imaging and Neuropixel recording data to interpret causal impacts of optogenetic suppression sampled across dorsal cortex. The authors present the novel findings that (1) effects of optogenetic suppression are specifically correlated with the magnitude of sensory encoding within each region and (2) frontal and occipital cortices make different contributions to choice formation (substrative vs additive).

We thank all three reviewers for their feedback on our paper. Following discussions with the Editor, we propose the following action plan. We start with Reviewer 3, who had the strongest concerns.Reviewer 3:

Reviewer 3 was concerned we cannot separate choice from action with our task design. We believe this concern arises from the terminology we used to describe our task, rather from our results themselves. In our task, the mouse has 3 options: to stay still (NoGo) or to turn the wheel one way (Go Left) or the other (Go Right). There is no standard terminology to describe the processes that adjudicate between these options, so following our previous paper (Steinmetz et al., Nature 2019) we defined the first adjudication (Go vs. NoGo) as 'action' and the second adjudication (Go Left vs. Go Right) as 'choice'. In retrospect this was problematic, and indeed it may have led to a misunderstanding with the reviewer: we should not have used words that are in generic use and imbue them with a specific meaning. We thus believe the reviewer's concern can be fixed by clarifying exactly what we meant to say, using different terminology.

First, let us restate our results without the words in question. Our data show that the differences in neural activity between Go and NoGo trials are of a fundamentally different character to the differences between Left and Right trials. Neurons whose activity differed between Go and NoGo trials were much more common across cortex than neurons distinguishing Left and Right trials, but areas with Go/NoGo correlates were not causally required for the task unless they also carry sensory information. Indeed, sensory encoding predicts causal relevance so well that we could quantitatively predict the effects of inactivation from sensory correlates alone.

Our use of "action" for the Go/NoGo adjudication and "choice" for the Go Left/Go Right adjudication was likely a mistake, because many readers likely use these words differently. Reviewer 3 commented that to separate action from choice the task "would need to have stimulus, delay, and response periods." We therefore believe that Reviewer 3 defines "choice" to mean a purely internal cognitive process occurring when the stimulus is first shown, and "action" to mean the execution of a physical movement. With this definition we agree that separating choice from action correlates would require a delay. The reviewer then cited several studies of Go/NoGo tasks (with no delay), noting that neural correlates of those outcomes (Go vs NoGo) are not sparse. Our data fully agree with these studies.

To avoid confusion, we therefore propose to adopt different terminology. Our plan is to call the neural differences between Go and NoGo "action initiation correlates" and the differences between Go Left and Go Right "action selection correlates". An alternative could be to call the first one "Go choice" and the second one "Left/Right choice". We are open to suggestions on this, as it is essential we use terminology that does not raise a risk of readers incorrectly interpreting our results. Another possibility is that we keep on calling them "action" and "choice" (consistent with Steinmetz et al. Nature 2019) but first giving an extensive definition (rather than a single sentence as we currently do).

Reviewer 3 raised several other important points, such as the possibility that neurons with action selection correlates might have specific projection targets, which we will address in the text. However s/he did not make specific suggestions for new analyses, and we therefore hope that carefully rewriting the paper with new terminology will address her/his concerns.

Reviewer 2 (Dr. Svoboda):

Reviewer 2 made several excellent suggestions, which we will address. The most important of these is to investigate inactivation in zero-contrast trials, for which we will include new figures and analyses. We have also thoroughly investigated the relationship between widefield and electrophysiological signals with regard to layers etc. This is described in a manuscript coming out in a few weeks (Peters et al., Nature in press), which we will cite. The reviewer made many other valuable points that we will address with changes to figures, analyses, and text.

Reviewer 4:

Reviewer 4 made several valuable suggestions, for example to emphasize more strongly the effect of MOp inactivation on peak velocity and to emphasize how these results differ to those of Steinmetz et al., Nature 2019. We will address these with textual edits.

[Editors’ note: what follows is the authors’ response to the second round of review.]

Reviewers made several positive comments about the ambitious nature of the study as well as the quality of the dataset, which combines many different techniques in a technically impressive manner. Reviewers generally commented positively on the identification of causally related task correlates related to sensory perception, which is based on a combination of optogenetics, recordings and modelling. However, they still had major concerns about the interpretation and analysis of the choice and action correlates, including about issues covered in the action plan that should be considered further, as follows:1) Explain conceptual advanceThe authors should better explain the novelty of their findings and the relationship to previous work (in particular Steinmetz et al. 2019). For example it needs to be clarified which aspects of findings using the same dataset are novel, e.g. the presentation of single unit choice probability.

All data are new except for the neuropixels recordings. Some analyses of the neuropixels data are new, for example stimulus decoding (Figure 1m), as well as the spatial mapping of vision, action, and choice correlates (1j-l) that allow comparison to widefield data.

We have changed the text to say “We compared the widefield results to previous analyses of Neuropixels electrode recordings made in the same task” (Line 120). This understates the novelty of the current study but we feel this is preferable to an exhaustive listing of exactly which analyses of the neuropixels data are novel, which would distract from our scientific conclusions.

2) Improve overall conceptual interpretation of task correlates and lack of choice correlates(i) The study should explain better how to understand the concept of choice in light of their task and findings. The authors should discuss why this is an appropriate task to study choice representation in the cortex, and how the interpretations of the authors might be constrained by not having separate stimulus, delay and response periods. The authors should improve their explanation of the difference between choice vs. no/go correlates.

The word “choice” may have different meanings to different readers. Because using this word in the previous version led to controversy, we feel it is better to avoid it altogether to avoid distracting from our scientific results.

We now use terms that we believe should be unambiguous. We say a neuron correlates with “vision” if it correlates with the sensory stimulus, even after accounting for behavior. We say a neuron correlates with “action selection” if it predicts whether the mouse will move left or right, even after accounting for the sensory stimulus. We say it correlates with “action execution” if it predicts that the mouse will move (in either direction), even after accounting for the sensory stimulus. These correlates can all be distinguished in our task, without requiring a delay period. In contrast, action selection and action execution could not be distinguished in a go/nogo task.

In this new language, our primary result is that the effect of silencing a cortical region correlates strongly with its encoding of vision, but not with its encoding of action execution. Cortical correlates of action selection were not even visible in widefield, and were found only in rare cells with no lateral bias. We have thoroughly revised the Discussion section to clarify the relationship of our results to previous studies (2AFC and go/nogo, with and without delays) (Lines 322 and subsequent paragraphs)

(ii) The authors should clarify the central of the logic of the study. If we consider the framework that decision-making occurs before action/response, then one would not expect action coding to impact 'a perceptual decision'. And yet, given the title, the authors appear to consider this a core finding of their study.

Again, we believe this should be cleared up by our new terminology, and have also changed the title to remove a possible source of confusion.

(iii) One argument weakening the conclusion about the finding that widespread action coding is not causal, is that the localized unilateral inactivation could be too weak to perturb distributed action coding. The authors point this out (Line 381), but it is hard to tell whether the current interpretation is "most parsimonious" (Line 385). Without bilateral and multi-regional inactivation data, the overall conclusion of this study should be more conservative. Instead of saying "do not play a causal role", descriptions like "not locally causal" (Line 415) may be more appropriate.

Agreed. We have tempered our claims and now use the term “locally causal” throughout the paper.

(iv) The authors should address the apparent discrepancy with studies (e.g. Lee at al, Cell Reports 2020; Tang et al. Neuron 2020; Puscian et al. Cell Reports 2020, etc) that show that neurons in primary visual cortex robustly encode choice in visual detection tasks.

Thank you for suggesting these references, which are entirely consistent with our results.

These papers used either a go/nogo task (Lee et al) or eyeblink conditioning (Tang et al; Puscian et al). A cell predicting behavior in these tasks therefore correlates with what we would term action execution (i.e. it fires preferentially before a go response or eyeblink) rather than with action selection (which would require it to fire differentially before two different actions).

We have rewritten the Discussion section to clarify this point (Lines 337 onwards)

(v) The authors should discuss the possibility that targeted recordings or imaging of specific populations according to projection patterns may reveal that choice encoding is robust, following the idea that there could be functionally separable components of large-scale circuits, and that choice encoding might a feature of neural populations with projection targets specific to the type of task.

Thank you, we have added a discussion of this (Lines 362 onwards).

3) Interpretation and analysis of action correlatesThe main conclusion of this study, as manifested in the title, is that the widespread cortical action coding is not causal. The authors should address several concerns in data analysis and result interpretation that may weaken this conclusion.(i) The conclusion is based on the negative results in Figure 3d-f, which averaged over equal non-zero contrast trials. Are these the best stimulus conditions to look at? It is reasonable to use these conditions in Figure 3a-c since the balanced left and right choices on control trials may maximize any Δ Contraversive effect of inactivation. However, following the same logic, it is more likely to observe significant ΔNoGo, if any, when the Go and NoGo choices are balanced, i.e., on trials with zero contrast on both sides (~50% NoGo in Figure 1f). In other words, using non-zero contrast trials, Figure 3d-f could have substantially underestimated the inactivation effect on Action.

Thank you for this suggestion, which we have added to Figure 3 supplement 1. On zero contrast trials there is no consistent effect of inactivating MOp or SSp. This is despite the fact that NoGo trials only occur with around 50% probability, so failure to increase this further cannot reflect a ceiling effect.

(ii) Also related to point (3-i), Figure 4d suggests that VIS or MOs inactivation does increase the NoGo choices for at least unilateral stimulus conditions. Although these two areas were assumed to be only sensory-related in the context of Figure 4, they also encode action (Figure 2). In particular, the pre-movement action coding in MOs is comparable with MOp (Figure 2n). Therefore, how to reconcile the observation in Figure 4d with the conclusion that cortical action coding is not causal?

Thank you for this point, which we again believe can be fixed with better terminology.

In our task, mice are rewarded for NoGo responses if no stimulus is presented. Our interpretation is that inactivating left VIS while presenting a stimulus on the right increases NoGo responses because VIS participates in routing visual information to downstream decision circuits; colloquially, the animal does not perceive the stimulus on the right if left VIS is inactivated, and makes the action that would result in a reward if no stimulus were present. The fact that inactivating left MOs has a similar effect suggests that MOs plays a similar role of routing sensory information to a downstream decision structure.

Our previous terminology, identifying some correlates as causal and others as non-causal, was inappropriate. We have changed the title, and the text throughout to clarify our new interpretation: the role of dorsal cortex in this task is to route sensory information to subcortical decision structures.

(iii) Related to the question whether action coding causally impact motor performance: Indeed, the authors report that MOp suppression reduced peak wheel velocity by 20%. The authors should discuss this finding more prominently in the Results and Discussion. This finding argue that the action signals in MOp do not solely reflect a corollary discharge. Did suppression in other 'action coding' regions also alter motor performance (wheel velocity, reaction time, other measures)?

Thank you for this suggestion, which we have added to Figure 3 Supplement 2e. Inactivating SSp did not have a consistent effect on wheel velocity, although inactivating MOs did.

We have also edited the text to more prominently state that MOp may be involved in action execution (lines 298 onwards).

4) Comparison between encoding vs. decoding results(i) In the encoding model, 20.1% of SSp neurons had significant pre-movement action encoding (Line 141), which was stronger than MOp (15.3%), whereas in Figure 2n, the action decoding of SSp is clearly weaker than MOp, with a proportion of significant pre-movement decoding far less than 20%. How can this discrepancy be explained?(ii) Line 56 "correlates of action.… strongest in primary motor and somatosensory cortex" and Line 390 "What then might be the function of the strong MOp/SSp activity observed prior to action execution?" Given the above discrepancy, how to define the strength of "action correlates"? By encoding or decoding? For example, based on decoding (Figure 2n), one would conclude that the strongest action correlates prior to movement is in MOp/MOs, or even VISal, but not in SSp.

The fraction of action-encoding cells did not differ significantly between SSp and MOp (p>0.05, Fisher’s exact test).

5) Comparison between modalitiesThe comparison between different methods, in particular widefield imaging and ephys recordings, should be explored better. Where do the widefield signals come from? How does it compare to the depth distribution sampled from Neuropixels recordings? Is there any laminar signature in the ephys data? For example, in Figure 2n, the SSp trace does not rise until -50 ms, but in the imaging data (Figure 2g), significant action signal already appears in SSp at -100 ms. Could this be explained by a sampling bias towards superficial layers in widefield imaging?

We have explored this relationship in a separate study (Peters et al. Nature 2021: extended data figure 4), which shows that widefield fluorescence correlates best with spiking in layer 5. We now cite this in the text (Lines 90, 364).

6) Use of CCCPCCCP, as a measure of choice probability, appears to be misleading when used to describe analyses to quantify sensory and action coding. One possibility is to introduce the general method of Combined Conditions Probability, and apply this analysis to Stimulus, Choice, or Action. (sCCP, cCCP, aCCP).

Thank you for the suggestion. We now simply refer to this as a ”binary decoder” method, which we expand on in the Methods (Line 105, 531)

7) Comparison between protocolsMany different protocols have been used throughout the paper, but the motivations are not always clear, making the results sometimes difficult to compare. For example:(i) The open-loop period was used in imaging and ephys sessions, but not in the inactivation experiments. Did the animals behave differently with and without the open-loop period?

We now show this in figure 3 supplement 2d. Open-loop mode prolonged the wheel movement, however the difference only appeared after the period on which we performed data analyses.

(ii) In the pulse inactivation experiment, 15 mW 25 ms photostimuli were used, but in Figure 3 Sup. 3c and f, the simultaneous ephys recordings were done with 4 mW, 10 ms. It is unclear how different these two protocols could be in terms of the long-lasting suppression effect, which is important for interpreting Figure 3h and i. Moreover, the rebound after 100 ms seems to be strong in Figure 3 Sup. 3c and f. Could it be even stronger in 15 mW 25 ms case? Rebound activity can greatly confound interpretation of inactivation experiments.

It is true that even a brief pulse can have long-lasting effects, including rebound. For example we see in Figure 3h that pulses delivered to visual cortex even before the stimulus appears can have an effect.

This is why when analyzing the data of figure 3h, the quantity we focused on was the latest time point at which a pulse had a significant effect. The fact that pulses delivered to VIS later than 130ms after stimulus onset do not have a significant effect indicates that VIS has completed its role in task performance by this time. This analysis is unaffected by the duration of the pulse inactivation effect, or of any rebound. We have clarified this in the text (lines 196 onwards).

(iii) In Figure 4, multiple laser powers were used, but the data were pooled together to compare with the neurometric model. Is this valid given that the effect of MOs inactivation depends on the laser power Figure 3 Sup.1f? This also raises a concern related to major point 3(3): what if higher powers are used in Figure 3d-f? It seems unfair to include 2.9 and 4.25 mW in Figure 4 while drawing conclusion from Figure 3d-f using only 1.5 mW.

We now provide an analysis of laser power in Figure 3 supplement 1f. MOp and SSp inactivation does not increase NoGo probability, regardless of laser power.

(iv) What is the reason for the additional optogenetic suppression experiments presented in in Figure 4 (how are they different from the experiments presented in Figure 3)? Are the lack of effects on NoGo probability in Figure 3D inconsistent with the increased Misses in Figure 4D? Is the lack of increased incorrect responses in MOs in Figure 4D inconsistent with the increased rightward choices in MOs in Figure 3A?

These follow-on experiments were done to allow multiple laser powers to be tested while focusing on a smaller number of inactivation sites. The data from figure 3 used one laser power and only bilateral visual stimuli. The data from figure 4 used 3 laser powers for bilateral, unilateral, and zero-contrast trials.

We have provided an expanded analysis of these data in Figure 3 supplement 1f, showing the effects of all powers on all stimuli. As can be seen in new this figure, 3D and 4D are not inconsistent: when bilateral visual stimuli are present, inactivating VIS or MOs replaces contralateral choices with ipsilateral choices; when only contralateral stimuli are present they are replaced by NoGos. We updated our explanation of this in the text (lines 170 onwards).

8) Neurometric model related to sensory correlates:The authors should discuss why the relatively simple neurometric model is appropriate. It uses one scalar variable to represent the population activity of each area, with an underlying assumption that all the areas are homogeneous. But this is not the case for MOs, where contra- and ipsi-lateral visual coding are mixed (Figure 2j and m). This may explain the strange result that the weights of MOs are additive (Figure 4b), which should be discussed. Is it possible to fit a similar neurometric model using the ephys data that at least takes this level of heterogeneity into account? The authors should address why a model with additive weights for MO and visual cortex is appropriate given that one might expect MO activity to depend on activity in the visual cortex.

We cannot fit a model based on ephys data unfortunately as this would require simultaneous recordings from VIS and MOs in both hemispheres, which we do not have.

MOs neurons are indeed heterogeneous, but their encoding of sensory stimuli is strongly biased towards the contralateral side: of the MOs neurons that encoded a visual stimulus, 80% of them encoded a contralateral stimulus. This does indeed provide a good explanation for why MOs have weights that are positive for both decisions, but stronger for the contralateral side. We have expanded discussion of these points in the text (Line 237, 267 onwards).